# 1 Simulating Precipitation-Induced Karst-Stream

# 2 Interactions Using a Coupled Darcy-Brinkman-Stokes

- Model
- Fuyun Huang<sup>1</sup>, Yuan Gao<sup>1\*</sup>, Zizhao Zhang<sup>1\*</sup>, Xiaonong Hu<sup>2</sup>, Xiaoguang Wang<sup>3,4</sup>,
- Shengyan Pu<sup>5</sup>
- School of Geology and Mining Engineering, Xinjiang University, Urumqi, Xinjiang 830046, China
- <sup>2</sup>School of Water Conservancy and Environment, University of Jinan, Jinan, Shandong 250022, China
- <sup>3</sup>State Key Laboratory of Oil and Gas Reservoir Geology and Exploitation, Chengdu University of
- Technology, Chengdu, Sichuan 610059, China
- <sup>4</sup>Tianfu Yongxing Laboratory, Chengdu, Sichuan 610059, China
- State Key Laboratory of Geohazard Prevention and Geoenvironment Protection, Chengdu University of
- Technology, Chengdu, Sichuan 610059, China
- Correspondence to: Yuan Gao (yuangao\_xju@hotmail.com) and Zizhao Zhang
- (zhangzizhao@xju.edu.cn)
- **Abstract.** The variation in seasonal precipitation intensity impacts the dynamic interaction between the 16 karst aquifer and stream. However, the interaction mechanism between the karst aquifer and stream is 17 currently still unclear, and characterizing the impact of dynamic saturation process of groundwater in 18 karst media on the interaction process remains a challenge. This study provides an in-depth analysis of 19 the interaction processes between karst aquifer systems and adjacent streams, along with water-air two-20 phase flow in aquifer media. Multiple water retention models were employed to characterize the soil-21 water characteristics of porous media and variably saturated groundwater flow. The research reveals that 22 rainfall intensity variations significantly influence the interactions between karst aquifer systems and 23 streams. These interactive processes become increasingly complex with higher rainfall intensities, 24 involving multi-media collaborative recharge and dynamic interactions, while the contribution 25 proportions of different media to streamflow also change accordingly. By comparing the modeling 26 differences and numerical results between CFPv2 and DBS approaches in generalized models, the 27 validity of the DBS model for groundwater modeling was verified. Under consecutive rainfall events,

total rainfall intensity plays a crucial role in hydrological process variations of adjacent streams. Groundwater stored in porous media of karst systems during the first rainfall event was found to influence stream water levels during subsequent rainfall events, while conduit storage exhibited minimal impact. Multi-level conduit configurations under specific conditions, particularly during intense rainfall, can significantly affect hydrological processes in both streams and karst conduits. Uncertainty analysis demonstrates that conduit geometry, diameter, epikarst permeability, and porosity differentially influence hydrological processes in karst aquifer systems. Variations in these parameters induce corresponding changes in peak flow rates, peak timing of stream and karst spring discharges, as well as redistribution of discharge contributions among different media, ultimately affecting the overall hydrological dynamics of the coupled karst aquifer-stream system. It can accurately depict the two-phase interactive flow between various media controlled by the dynamic saturation process, and reveal the dynamic interaction process between karst aquifers affected by the epikarst, sinkholes, and conduits under infiltration recharge and stream. Meanwhile, it can precisely explain the processes of infiltration, overflow, and recession.

- **Keywords:** the karst aquifer and stream; precipitation recharge; two-phase flow; Darcy-Brinkman-
- 43 Stokes equation; interaction mechanism

# 1 Introduction

Karst aquifer is not only a repository of substantial freshwater resources (Li et al., 2017; Ford & Williams, 2007; Sivelle et al., 2021), but also provides drinking water for 10% to 25% of the global population (Longenecker et al., 2017; Goldscheider et al., 2020; Mahler et al., 2021). However, karst-developed areas feature intricate pore structures and fractures (Kuniansky, 2016), leading to pronounced heterogeneity and anisotropy in the movement and storage of water within them (Zhang et al., 2020). In particular, the complex coupled flow involving various flow paths such as karst conduits, sinkholes, and epikarst, along with porous media, further intensifies the nonlinear recharge and discharge processes and the formation of preferential flow paths in the karst aquifer. With seasonal variations in precipitation intensity, the heterogeneity of the groundwater flow field is further exacerbated, and water levels in the karst aquifer and stream fluctuate, leading to complex interactions between the aquifer and stream

(Bonacci, 2015). Unveiling the interaction mechanism between the karst aquifer and stream under varying precipitation intensities is crucial for assessing the storage of water resources in karst regions (Gao et al., 2021; Guo and Jiang, 2020).

The interaction process between the karst aquifer and stream is significantly influenced by karst media. In epikarst where the soil layer is shallow and dissolution weathering is pronounced, most precipitation can directly recharge the karst aquifer (Lee and Krothe, 2001; OLello et al., 2018). Karst conduits and sinkholes are important media involved in karst hydrological cycle. As rapid discharge channels, the size, connectivity, and distribution of karst conduits have a significant impact on karst hydrological processes (Duran et al., 2020; Bittner et al., 2020). Surface water collected into sinkholes can directly recharge the karst aquifer (Bianchini et al., 2022), thereby regulating the water level of the aquifer and the discharge volume to the stream, which is influenced by precipitation intensity, size and distribution of sinkhole. The permeability of sinkholes and conduits typically exhibits multilevel characteristics and varies with scale (Halihan et al., 1999), meaning there are strata structures with different permeabilities, which complicates the flow of water within the karst aquifer and increases the catchment area.

Numerical methods are commonly employed as effective means to accurately simulate karst groundwater movement and assess karst groundwater resources. Shoemaker et al. (2008) proposed a method that discretely embeds conduits, connected by nodes, into the porous media grid (MODFLOW-CFP). This method not only evaluates the water resources of the entire karst aquifer but also considers the geometric shape and distribution of karst conduits on the hydrological processes. Moreover, this methodology has been extensively applied worldwide for estimating karst groundwater flow and water resources (Chang et al., 2015; Qiu et al., 2019; Kavousi et al., 2020; Gao et al., 2020, 2024), as well as in integrated modeling studies coupling SWAT with MODFLOW to investigate groundwater-surface water interactions (Fiorese et al., 2025; Yifru et al., 2024). While MODFLOW-CFP provides robust capabilities for regional-scale karst groundwater simulations, it currently supports only single-phase groundwater flow modeling. Although MODFLOW-CFP is relatively comprehensive for regional karst groundwater simulation studies, the current version of MODFLOW-CFP only supports modeling single-phase groundwater flow.

The interaction process between the karst aquifer and stream is also regulated by the dynamic saturation process within the aquifer. The degree of dynamic saturation in different media determines the path and velocity of water flow. Unsaturated aquifers gradually saturate the underlying aquifers under the influence of gravity, while saturated underlying aquifers can cause water to overflow (Worthington, 1991; Huang et al., 2024). In addition, the dynamic saturation processes within the karst aquifer are regulated by factors such as seasonal water level fluctuations, the infiltration and flow of groundwater, and the periodic filling and draining of karst conduits (Huang et al., 2024).it is necessary to couple seepage (porous media) with free flow (conduits and stream) and to describe the dynamic saturation process of the karst aquifer. The Hydrus simulation method based on the Richards equation is capable of simulating variably saturated flow (Dam and Feddes, 2000). However, this approach lacks a built-in conduit flow solution scheme, making it difficult to adequately address the coupling requirements between rapid conduit flow and porous media seepage in karst areas.

Constructing an interaction model between the karst aquifer system and the stream under rainfall event-driven conditions requires coupling free flow and seepage processes while simultaneously supporting two-phase variably saturated flow. (1) The Navier-Stokes (N-S) equations combined with the Darcy equation can effectively couple free flow and seepage processes (Soulaine and Tchelepi, 2016; Carrillo et al., 2020).(2) The Phase Indicator Function for two-phase flow, combined with the phase transition method, can effectively describe the variable saturation process within the karst aquifer (Huang et al., 2024; Zhai et al., 2024). The Darcy-Brinkman-Stokes equations have been utilized to couple seepage flow and free flow (Huang et al., 2024; Nillama et al., 2022; Carrillo et al., 2020). Lu et al. (2023) analyzed a model that integrates fast discharge channels in fractures and conduits with slow seepage in porous media. The results demonstrate that the Darcy-Brinkman-Stokes equations can effectively describe two-phase flow in karst aquifers, and Soulaine (2024) proposed that mixed-scale models based on the Darcy-Brinkman-Stokes equations have strong potential for simulating coupled processes in porous systems.

The karst aquifer are typically accompanied by turbulent flow. Reimann et al. (2011) conducted thorough research on turbulent flow in the karst aquifer. To reflect the dissipation of turbulent processes throughout the system, the N-S and Darcy-Brinkman-Stokes equations can be studied using the Reynolds

Averaged Network System (RANS) method, where the k-ε turbulence model effectively characterizes turbulent flows in porous media, as demonstrated by del Jesus et al. (2012). The RANS method has been progressively refined for evaluating turbulent flow in both free-flow regions and porous media (Huang et al., 2024; Zhai et al., 2024; Higuera et al., 2014).

This study aims to employ a two-phase variably saturated model capable of coupling free flow and seepage flow to reveal the interaction mechanisms between the karst aquifer system and adjacent stream under rainfall infiltration recharge-driven conditions. Specifically, it focuses on further investigating how groundwater saturation variations in different media (e.g., conduits, fractures, matrix) of the karst aquifer system influence inter-media interactions. This research addresses the gap in existing studies where current numerical methods struggle to accurately characterize the collaborative recharge processes among various media within karst aquifer systems. This study employs the Darcy-Brinkman-Stokes equations to model the coupled processes of seepage in porous media and free flow in karst conduit and stream. The Brooks-Corey (BC) and van Genuchten-Mualem (VGM) models are used to characterize the unsaturated seepage in karst media. The Volume of Fluid (VOF) method is applied to monitor the dynamic changes in aquifer saturation. This research elucidates how saturation dynamics in different karst media impact the coordinated recharge among media during precipitation infiltration, and examines the evolving interaction between the karst aquifer and stream under such recharge conditions. Given the complexity of the interaction mechanism between the karst aquifer and stream, this study specifically investigates the impact of four factors on the interaction mechanism: (1) changes in precipitation intensity, (2) different water retention models, (3) multi-stage conduit arrangements, and (4) parameter sensitivity analysis. The research results can further reveal the interaction mechanisms between karst systems and adjacent streams under rainfall infiltration recharge, and provide an in-depth discussion on the impacts of rapid seepage, overflow, and sudden changes in spring discharge on flood control and overflow management along the stream. This study offers a scientific basis for accurately and rationally assessing karst water resources.

### 2 Materials and methods

To quantitatively characterize the interaction processes between karst aquifer systems and adjacent

streams, as well as the groundwater flow through various media within karst aquifers, the Darcy-Brinkman-Stokes (DBS) method was employed to couple seepage and free flow. The Volume of Fluid (VOF) method was applied to characterize water-air two-phase flow in heterogeneous media, while different water retention models were implemented to describe unsaturated flow processes in karst groundwater systems.

# 2.1 Numerical modelling

The numerical model is developed according to the conceptual model of the karst aquifer adjacent to a stream, as depicted in Fig. 1. The model construction incorporates distinct rainfall intensities and temporal rainfall patterns (Figure 1(a)-(b)), while explicitly accounting for characteristic karst geomorphological features including sinkholes, epikarst, and karst conduits. The karst conduit is connected to the epikarst through a sinkhole. The outcrop of the karst spring is located at the end of the karst conduit, directly leading to the stream.

Recharge Pathways in a Single Recharge Event: During a single recharge event, precipitation follows two main pathways: a portion directly recharges the adjacent stream, while another portion infiltrates into the epikarst zone (shallow karst system). A fraction of the water stored in the epikarst zone discharges laterally to the stream, while the remaining water disperses vertically through porous media to recharge the deeper porous aquifer. The residual water in the epikarst zone further recharges the karst conduit system via sinkhole point infiltration (Figure 1(a.1)).

Conduit Network-Matrix Interaction: Under moderate recharge events, conduits receive water from both sinkhole point recharge and porous media recharge, rapidly transporting it to discharge at karst springs. During intense precipitation events, water in the conduits may temporarily reverse flow to recharge the porous media before returning to the conduits (Bailly-Comte et al., 2010).

Karst Aquifer-Stream Interaction: Lateral recharge from the porous aquifer to the stream requires prior vertical dispersion recharge from the overlying epikarst zone. During a single precipitation event, direct lateral recharge from the epikarst zone and rapid discharge of groundwater from karst springs to the stream cause an earlier stream stage rise. As the stream stage gradually increases, the stream begins to recharge the deeper porous media of the karst aquifer (Figure 1(a.1)). Due to the high flow velocity of the stream, its stage declines rapidly, allowing groundwater in the deeper porous media to discharge back

into the stream (Figure 1(a.2)).

The precipitation influences the dynamic variation process of saturation within porous media, and the water levels in both the karst conduit and the stream experience substantial fluctuations. As a result, the interaction between the porous media and the stream displays a clear multi-scale characteristic. From a hydrological perspective of the watershed, the recharge and discharge processes of karst conduit are controlled by the saturation degree of the surrounding porous media and the water level within the conduit themselves. Based on spatial relationships, the area between the karst conduit and the epikarst is divided into Porous Medium II (PM II) above the conduit, Porous Medium II (PM III) on the sides, and Porous Medium III (PM III) directly below the conduit(Figure 1(a.1)). Based on the aforementioned dynamic interaction processes between the karst aquifer system and the adjacent stream, this study constructs the DBS numerical model and employs the CFPv2 (Shoemaker et al., 2008; Giese et al., 2018) to simulate groundwater flow. Through analyses of precipitation intensity variations, multiple precipitation events, different water retention models, multi-level permeability configurations, and parameter sensitivity analyses under repeated rainfall influences, the interaction mechanisms between the karst aquifer system and the stream are elucidated.

# 2.2 DBS model

#### 2.2.1 Two-Phase Flow Parameter Definition

Assuming that gas and liquid fill the solid pore space, porosity is defined to characterize the percentage of the gas and liquid phases occupying the total pore space.

$$\varphi = \frac{V_l + V_g}{V} \tag{1}$$

In this context,  $\varphi$  represents porosity, V denotes the total volume of the unit [m<sup>3</sup>], while  $V_l$  and  $V_g$  correspond to the volumes of the liquid phase (water) and gas phase (air), respectively [m<sup>3</sup>].

Hirt and Nichols (1981) introduced the Volume of Fluid (VOF) method, which employs an additional governing equation to capture fluid motion at free surfaces. Furthermore, the saturation of each phase in the fluid is defined as  $\alpha_i$ , where:

191 Liquid phase saturation: 
$$\alpha_l = \frac{v_l}{v_g + v_l}$$
.

- Gas phase saturation:  $\alpha_g = \frac{V_g}{V_g + V_l}$ .
- Here, the subscripts *l* and *g* denote water and air, respectively. Thus, the spatial distribution of water
- and gas within the porous medium is characterized by porosity  $\varphi$  and phase saturation  $\alpha_i$ :

$$\varphi = \begin{cases} 1 & \text{free regions} \\ 0 < \alpha < 1 & \text{porous regions} \\ 0 & \text{solid regions} \end{cases}$$
 (2)

$$\alpha_l = \begin{cases} 1 & \text{water} \\ 0 < \alpha < 1 & \text{two-phase zone} \\ 0 & \text{air} \end{cases}$$
 (3)

The average fluid density  $\rho[m^3/kg]$  and viscosity  $\mu[m^2/s]$  within a grid cell are calculated via saturation-weighted averaging:

$$\rho = \rho_g \alpha_g + \rho_l \alpha_l \tag{4}$$

$$\mu = \alpha_g \mu_g + \alpha_l \mu_l \tag{5}$$

- where  $\rho_g$  is the gas phase density [m<sup>3</sup> /kg] and  $\rho_l$  is the liquid phase (water) density [m<sup>3</sup> /kg].
- The transport equation for saturation  $\alpha_i$ , following Rusche (2002), is expressed as:

$$\frac{\partial \varphi \alpha_l}{\partial t} + \nabla \cdot (\alpha_l v_t) + \nabla \cdot (\varphi \alpha_l \alpha_g v_{rt}) = 0 \tag{6}$$

- where:  $v_t$  is the fluid velocity vector  $[m/s], v_{rt}$  is the relative velocity between the gas and liquid
- phases [m/s].

# 206 2.2.2 Governing Equations

To precisely describe groundwater flow through porous media in the karst aquifer system and the free-surface flow processes between conduits and the adjacent stream, this study adopts the DBS (Dualdomain Brinkman-Stokes) equations to characterize immiscible and incompressible two-phase flow in porous media (Nillama et al., 2022; Carrillo et al., 2020; Lu et al., 2023; Huang et al., 2024; Soulaine, 2024). The DBS model is employed to represent both Darcian flow in porous media and turbulent flow dynamics during free-surface interactions between conduits and the stream. The governing equations include:

$$\nabla \cdot \overline{v} = 0 \tag{7}$$

$$\frac{\partial \varphi \alpha_l}{\partial t} + \nabla \cdot (\alpha_l \overline{v}) + \nabla \cdot (\varphi \alpha_l \alpha_g \overline{v_r}) = 0 \tag{8}$$

$$\frac{1}{\varphi} \left( \frac{\partial \rho \overline{v}}{\partial t} + \nabla \cdot \left( \frac{\rho}{\varphi} \overline{v} \overline{v} \right) \right) = -\nabla \bar{p} + \rho g + \nabla \cdot \left( \frac{\mu}{\varphi} \left( \nabla \overline{v} + \nabla \overline{v}^T \right) \right) + F_c + S_f. \tag{9}$$

- Here, t represents the computational time [T],  $\overline{v}$  is the velocity [L/T],  $\overline{v_r}$  is the relative flow rate
- of the gas phase to the liquid phase [L/T],  $\rho$  is the average density of the gas and liquid phases  $[M/L^3]$ ,
- $\bar{p}$  is the pressure [pa], g is is the acceleration due to gravity (9.81 m/s<sup>2</sup>),  $\mu$  is the viscosity [L<sup>2</sup>/T],  $F_c$  is
- the surface tension, and  $S_f$  is the resistance source term.
- Conduit networks in karst aquifer systems are often associated with turbulent flow (Reimann et al.,
- 2011). To resolve turbulence in the DBS (Dual-domain Brinkman-Stokes) equations, the Unsteady
- Reynolds-Averaged Navier-Stokes (URANS) framework is required. As demonstrated by del Jesus et al.
- (2012), the k-epsilon turbulence model is effective for evaluating turbulent processes within porous
- media. Consequently, the k-epsilon-based DBS turbulence governing equations are formulated as follows:

$$\nabla \cdot v_t = 0 \tag{10}$$

$$\frac{\partial \varphi \alpha_l}{\partial t} + \nabla \cdot (\alpha_l v_t) + \nabla \cdot (\varphi \alpha_l \alpha_g v_{rt}) = 0 \tag{11}$$

$$\frac{1}{\varphi} \left( (1+c) \frac{\partial \rho v_t}{\partial t} + \nabla \cdot \left( \frac{\rho}{\varphi} v_t v_t \right) \right) = -\nabla p^* + \rho g \cdot X + \nabla \cdot \left( \mu_{eff} (\nabla v_t + \nabla v_t^T) \right) - \mu_{eff} k^{-1} v_t + F_c.$$
 (12)

- where,  $v_t$  represents the turbulent velocity vector [L/T],  $v_{rt}$  is the relative velocity of gas-phase and
- water-phase turbulence [L/T], and  $\mu_{eff}$  is the effective viscosity, which can be defined as  $\mu_{eff} = \mu +$
- $\rho v_{turb}$ , where  $\mu$  is the dynamic viscosity and  $v_{turb}$  is the turbulent kinetic viscosity.
- The eddy viscosity is expressed as:

$$\mu_t = \rho C_\mu \frac{k_t^2}{\varepsilon} \tag{13}$$

- where:  $k_t$ : Turbulent kinetic energy per unit mass  $[m^2/s^2]$ ,  $\varepsilon$ : Turbulent dissipation rate per unit
- mass  $[m^2/s^3]$ ,  $C_{\mu}$ : Dimensionless constant with a value of 0.09.

### 2.2.3 Subdomain Formulation

For the free-flow region and the porous media region, the source terms in the DBS equations adopt

- distinct forms. Specifically, the source term  $\mu k^{-1}$  in the two regions can be expressed as (Soulaine, 2024;
- Huang et al., 2024):

$$\mu_{eff}k^{-1} = \rho v_{turb}k^{-1} + \begin{cases} 0, & \text{free region} \\ k_0^{-1} \left(\frac{k_{r,l}}{\mu_l} + \frac{k_{r,g}}{\mu_g}\right)^{-1}, & \text{porous region} \end{cases}$$
(14)

- Here,  $k_0$  is the permeability coefficient determined by the pore structure [ $m^2$ ]. When the
- permeability is extremely high, this term vanishes, and the DBS equations reduce to the Navier-Stokes
- (N-S) equations (Equation 15). Conversely, as permeability decreases, the term  $\mu k^{-1} \overline{\nu}$  becomes
- dominant compared to other source terms, causing the DBS equations to asymptotically approach the
- Darcy equation incorporating gravity and surface tension (Equation 16).

$$(1+c)\frac{\partial \rho v_t}{\partial t} + \nabla \cdot (\rho v_t v_t) =$$

$$-\nabla p^* + \rho g \cdot X + \nabla \cdot \left(\mu_{eff}(\nabla v_t + \nabla v_t^T)\right) + F_{c,i}f\varphi = 1.$$
(15)

$$0 = -\nabla p^* + \rho g \cdot X - \mu_{eff} k^{-1} v_t + F_c, if \varphi \in ]0,1[.$$
 (16)

- Similarly, the surface tension force  $F_c$  and density  $\rho$  in the two regions can be expressed as (Huang
- et al., 2024):

$$F_{c} = \begin{cases} -\frac{\sigma}{\varphi} \nabla \cdot \left(\frac{\nabla \alpha_{l}}{|\nabla \alpha_{l}|}\right) \nabla \alpha_{l}, & \text{free region} \\ \left[k_{0} \frac{\left(\frac{k_{r,l}}{\mu_{l}} \alpha_{g} - \frac{k_{r,g}}{\mu_{g}} \alpha_{l}\right)}{\frac{k_{r,l}}{\mu_{l}} + \frac{k_{r,g}}{\mu_{g}}} \left(\frac{\partial p_{c}}{\partial \alpha_{l}}\right) - p_{c}\right] \nabla \alpha_{l}, & \text{porous region} \end{cases}$$

$$(17)$$

$$\rho = \begin{cases}
\rho_{l}\alpha_{l} + \rho_{g}\alpha_{g}, & \text{free regions} \\
\left(\frac{\rho_{g}\frac{k_{r,g}}{\mu_{g}} + \rho_{l}\frac{k_{r,l}}{\mu_{l}}}{\frac{k_{r,l}}{\mu_{l}} + \frac{k_{r,g}}{\mu_{g}}}, & \text{porous regions} \end{cases}$$
(18)

- Here,  $\sigma$  is the interfacial tension [N/m],  $p_c$  is the capillary pressure [pa], and  $k_{r,g}$
- and  $k_{r,l}$  represent the relative permeabilities of the gas phase and liquid phase, respectively.

#### 2.2.4 Relative Permeability Model

- Accurate modeling of two-phase flow in porous media is critical in geosciences. Simulating two-
- phase flow in variably saturated porous media requires precise estimation of the relationship between
- relative permeability and saturation (Springer et al., 1995).

To characterize the variation in two-phase relative permeability, the effective saturation of the liquid phase must first be defined. This is expressed as:

$$\alpha_{l,e} = \frac{\alpha_l - \alpha_{l,r}}{1 - \alpha_{q,r} - \alpha_{l,r}} \tag{19}$$

- where:  $\alpha_{l,e}$  denotes the effective water saturation,  $\alpha_l$  and  $\alpha_{l,r}$  represent the water saturation and residual water saturation, respectively, and  $\alpha_{g,r}$  is the residual air saturation.
- Relative permeability is a critical parameter in groundwater and related engineering fields (Kuang and Jiao, 2011). The Brooks and Corey (BC) model (Brooks and Corey, 1964) and the van Genuchten model (van Genuchten, 1980) are widely used as representative relative permeability models. The BC model establishes a relationship between relative permeability and effective water saturation as follows:

$$k_{rg} = (1 - \alpha_{l,e})^n \tag{20}$$

$$k_{rl} = \alpha_{l,e}^n \tag{21}$$

 $k_r$  denotes the relative permeability, where n is a dimensionless coefficient determined by the properties of the porous medium. The Brooks-Corey (BC) model exhibits a sharp discontinuity at the air entry point, which can lead to poor data fitting, particularly for fine-textured soils (Assouline & Or, 2013). The van Genuchten (1980) model addresses this limitation. By incorporating the parameter m = 1 - 1/n proposed by Mualem (1976), the modified van Genuchten-Mualem (VGM) model (Parker et al., 1987) is formulated as:

$$k_{r,g} = \left(1 - \alpha_{l,e}\right)^{0.5} \left(1 - \alpha_{l,e}^{1/m}\right)^{2m} \tag{22}$$

$$k_{r,l} = \alpha_{l,e}^{0.5} \left( 1 - \left( 1 - \alpha_{l,e}^{1/m} \right)^m \right)^2 \tag{23}$$

Here, m is a dimensionless parameter.

The selection of permeability equations is critical for appropriate predictions of relative permeability (Yang et al., 2019), indicating that pore tortuosity-connectivity plays a dominant role in groundwater two-phase flow. Therefore, this study conducts simulations and parameter sensitivity analyses for both the Brooks-Corey (BC) and van Genuchten-Mualem (VGM) models.

# 2.3 CFPv2 model

The CFPv2 model, proposed by Reimann et al. (2014), is an advanced version of MODFLOW-CFP (Shoemaker et al., 2008). It extends functionalities such as flow interactions between conduits and porous media, as well as conduit boundary conditions. CFPv2 integrates with MODFLOW-2005 and employs the following approaches: Laminar Flow in Conduits: Described using the Hagen-Poiseuille equation for discrete conduits within conduit networks. Turbulent Flow: Calculated by combining the Darcy-Weisbach equation with the Colebrook-White equation. Laminar Flow in Fractured Rock Matrix: Simulated via a continuum approach. Detailed technical documentation for MODFLOW-CFP, including groundwater flow simulation methodologies, is provided by Shoemaker et al. (2008). Successful applications and evaluations of the model have been reported in studies such as Gallegos et al. (2013), Reimann et al. (2014), Chang et al. (2019), Gao et al. (2020), and Shirafkan et al. (2023).

#### 2.4 Model Comparison and Numerical Model Construction

#### 2.4.1 DBS Model Conversion and Applicability Assessment

As illustrated in Figure 2, the Navier-Stokes (N-S) model can resolve fine-scale pore-scale flows and perform high-fidelity simulations. In contrast, the CFPv2 model achieves high computational efficiency and stability by discretizing one-dimensional conduits within porous media. The DBS (Dual-domain Brinkman-Stokes) model combines the advantages of both approaches: By incorporating additional resistance source terms into the N-S equations, it maintains high-fidelity flow resolution in conduits. For porous media, it adopts a Darcy-type flow formulation, significantly reducing computational costs.

However, the DBS model operates in three dimensions (3D), requiring grid refinement around conduits and their vicinity to ensure accurate flow resolution. This increases computational load compared to the 1D conduit flow framework of CFPv2. To address this challenge, all simulations in this study were executed on a Lenovo ThinkSystem SR665 server, which provides the necessary computational power for handling complex 3D meshes.

#### 2.4.2 Model Comparison and Discretization Schemes

To further investigate the effectiveness of the DBS model in addressing interactions between karst groundwater and adjacent streams, this study compares the differences between the MODFLOW-CFP and DBS models. As shown in Figure 3(a.1), the comparison begins with their coupling modes of conduits and porous media from the perspectives of governing equations and grid discretization: MODFLOW-CFP: Groundwater exchange between conduits, porous media, and streams relies on stable hydraulic heads between conduit-porous media and stream-porous media interfaces (Figure 3(a.2)). Flow interactions between porous matrix and discrete conduits are linear and driven by head differences (Barenblatt et al., 1960). DBS Model: Groundwater interactions among conduits, streams, and porous media are governed by saturation and pressure gradients between adjacent grid nodes, allowing simultaneous recharge or discharge across interfaces (Figure 3(a.3)). However, this requires calculating flux variations across all grids. Comparison of Stream-Porous Media Interaction Modes: MODFLOW-CFP: Streams are discretized into single grid cells, with exchange fluxes determined by head differences. Fluctuating stream stages are simplified to a uniform water level, and "dry zones" cannot be simulated in porous media (Figure 3(a.4)). DBS Model: Media properties (e.g., porosity, permeability) are assigned at grid nodes, and interface values are interpolated. Direct conduit-stream interactions eliminate the need for porous media as an intermediary. Stream geometry can be defined as regular (rectangular) or irregular (Figure 3(a.5)). The DBS model employs the Volume of Fluid (VOF) and Front-tracking methods to reconstruct dynamic water-air interfaces, enabling simulation of fluctuating interfaces under sufficiently refined grids. Discretization Schemes: This study adopts a dynamic programming approach to generate sinkhole and conduit grids, allowing flexible placement of conduits with adjustable diameters and coordinates, enhancing model adaptability (contrasting fixed conduit positioning in studies like Kavousi et al., 2020; Pardo-Igúzquiza et al., 2018; Li et al., 2023). DBS Discretization (Figure 3(b)): The epikarst layer thickness and stream location are defined. Regions are divided into free-flow zones (streams, sinkholes, conduits) and porous media. Free-

flow zones use locally refined grids to capture micro-scale variations in water levels and interfaces.

Porous media zones adopt gradually coarsening grids (edge cells twice the size of conduit-adjacent cells), balancing accuracy and computational efficiency. Permeability is graded, decreasing outward from conduits to reflect dissolution effects.

CFPv2 Discretization (Figure 3(c)): Conduits are embedded in porous media and directly connected to streams. Domain dimensions:  $200 \text{ m} \times 200 \text{ m} \times 30 \text{ m}$  (length  $\times$  width  $\times$  thickness). Groundwater flows from porous media to conduits and discharges into streams (Figure 11(a.1)).

Porous media: Homogeneous, initial head =10 m, no-flow boundaries. Conduits: Diameter =1 m, roughness =0.01 m, wall interaction parameter =25 m/s, outlet collocated with stream grid. Initial conditions: Spring discharge =0, conduit node elevation =1 m, water temperature =20°C. Boundary conditions: Rainfall recharge at the top, total simulation time =45,000 s, MODFLOW-CFP stress periods =1 min.

# 2.5 Rainfall Infiltration Recharge Boundary

The upper boundaries of both the DBS and CFPv2 models are defined as transient natural precipitation boundary conditions. In this study, the rainfall infiltration recharge boundary condition is formulated as follows (Huang et al., 2024; Chang et al., 2015):

$$I(t) = \frac{b}{\sqrt{2\pi\sigma^2}} \sum_{i} e^{-\frac{\left(\frac{t_i - \mu}{a}\right)^2}{2\sigma^2}}$$
 (24)

Here,  $t_i$  denotes the time of the i-th rainfall event, and I(t) represents the total rainfall amount at that time. According to Chang et al. (2015), the parameters  $\mu$ ,  $\sigma^2$ , and a are set as constants (90, 1.5, and 20, respectively). Variations in rainfall intensity during the infiltration recharge process are controlled by adjusting the value of the dimensionless parameter b.

# 3 Results

3.1 Interaction process between the karst aquifer and stream under precipitation infiltration recharge

## 3.1.1 Karst Aquifer-Stream Interactions Under Varying Precipitation Intensities

The changes in hydrological process curves, water level fluctuations, and their differences during the interaction between karst media and stream under different precipitation intensities

are shown in Fig. 4. In the early stage of precipitation, the flow in the stream primarily originates from direct precipitation recharge and lateral groundwater recharge from epikarst (Fig. 4(a)). As the water level in the stream gradually rises, the flow not only continues downstream but also begins to recharge the karst aquifer, particularly the PM II. The peak recharge to PM II coincides with the peaks of epikarst recharge to the stream (Epikarst in Fig. 4) and direct precipitation recharge (P-River in Fig. 4). Therefore, the interaction process between the karst aquifer and stream during the early precipitation stage is significantly influenced by lateral groundwater discharge from the epikarst and the direct precipitation recharge. As groundwater recharge from epikarst to the stream declines (Fig. 4 (a)), groundwater moves downward through the epikarst to PM I, and begins to gradually recharge the stream. Due to the low permeability of the epikarst, lateral discharge from PM I to the stream will be delayed. During this process, the discharge volume of PM I exhibits two distinct peaks. The first peak is due to the recharge of groundwater from the epikarst, while the second peak is caused by the gradual saturation of PM II and the karst conduit, with a proportion of groundwater overflowing from PM I and discharging laterally to the stream. After the end of precipitation recharge, the hydrological process curve of PM I rapidly declined, and the discharge volume of the karst conduit, PM III and PM II gradually increase, causing the water level in the stream to rise (Fig. 4 (d)). When the water level in the stream gradually exceeds that of PM I, the stream begins to gradually recharge PM I. The karst conduit, PM II and PM III continue to discharge to the stream during this stage due to higher internal water pressure, forming a local hydrological cycle with the upper layer. In the late stage of precipitation, the hydrological process of the stream primarily shows a gradual decline in baseflow.

As depicted in Figs. 3b and 3c, the recharge and discharge dynamics between the karst aquifer and stream across different media shift notably with escalating precipitation intensity. The recharge volumes from the stream to PM I and PM II both decrease. The reduction in the recharge to PM II from the stream is primarily due to the acceleration of groundwater movement downward as precipitation intensity increases, causing groundwater to move more rapidly to the bottom of the karst aquifer, thereby recharging PM II. Consequently, part of pore space that should have been recharged by the stream is instead recharged from PM I downward. The decrease in the

recharge to PM I can be attributed to its high internal saturation level and the rise in water level. On the other hand, the water level in the stream does not significantly exceed that of the upper aquifer, making it difficult for the stream to effectively recharge the aquifer. Due to the reduced recharge volume to the aquifer, the discharge from the stream is partially lower than the discharge from the epikarst during the early stage of the hydrological process.

With changes in precipitation intensity (b = 3, 5, and 7), the water level variations and their differences between the karst aquifer and stream exhibit complex dynamic characteristics (Figs. 3d, 3e and 3f). During the early stage of precipitation, despite the increasing water level difference, the discharge from the stream to the aquifer is gradually decreasing (as shown by the negative values for PM I and PM II in Fig. 4a, 3b and 3c). This phenomenon indicates that water level is not the only factor controlling the interaction between the karst aquifer and stream; changes in the degree of saturation also play a significant role. As shown in Fig. 4d, under low precipitation intensity, the water level difference between the karst aquifer and stream is often greater than the water level of the stream during the middle and later stages of precipitation. However, as precipitation intensity increases, the water level difference tends to decrease (Fig. 4b and 3c). This change is primarily due to the increased precipitation intensity leading to a faster saturation of the karst aquifer, thereby limiting the ability of the stream to recharge the aquifer. After the middle stage of precipitation, the interaction between the stream and the upper part of the aquifer gradually intensifies, while the lower part of the aquifer discharges to the stream (Fig. 4a). Due to the gradual decrease in water level difference, it is difficult for the stream to effectively recharge the aquifer. In this process, the interaction between the aquifer and stream is controlled by the dynamic changes in saturation.

Based on the comparison between DBS and Modflow-CFPv2 results in Figs 4(a), (b), and (c), the CFPv2 model exhibits a single-peak hydrograph with exponential recession characteristics, failing to capture flow process line disturbances caused by multi-media interactions. Under precipitation intensities b=3 and 5, the CFPv2 model shows an immediate rapid increase in stream discharge during early stages rather than gradual enhancement, though total discharge and baseflow during later stages remain comparable (as shown in Table 3). Specifically, for b=3, the

peak stream discharge in Modflow-CFPv2 occurs at 2520 s, earlier than in the DBS model. This discrepancy arises because the precipitation recharge package in CFPv2 directly elevates water levels, whereas the DBS model simulates a gradual vertical infiltration process along the Z-axis. Lower precipitation intensity reduces groundwater infiltration rates and prolongs water table replenishment time, consequently delaying lateral discharge timing. At b=7, both models exhibit comparable first discharge peaks, but the DBS model generates a secondary peak through overflow effects that rapidly recedes after overflow cessation. In contrast, CFPv2 demonstrates smooth exponential recession without secondary features due to its simplified vertical stratification that neglects multi-component interactions.

The comparable results between DBS and Modflow-CFPv2 models under variable recharge conditions demonstrate the reliability and stability of DBS in simulating karst aquifer systems. Although the DBS model captures more interaction details, it requires greater computational resources. The absence of overflow mechanisms and multi-media interactions in CFPv2 leads to simplified discharge recession patterns that fail to reflect intense component interactions within the system. This comparative analysis highlights the DBS model's advantages in characterizing complex conduit-stream-aquifer interactions while acknowledging its computational demands.

It is self-evident that changes in precipitation intensity significantly affect the recharge and discharge processes between the karst aquifer and stream. The water levels and saturation degrees of the respective media act as core controlling factors that jointly influence the interactive dynamics between the aquifer and stream. To gain a deeper understanding of these influencing factors and their interaction mechanisms, and to further elucidate the interaction process mechanisms between the karst aquifer and stream, this study focuses on the hydrological interaction process between the two during the early stage of precipitation.

### 3.1.2 Interaction process between the karst aquifer and stream during early stage of precipitation

Figure 5 illustrates how the interaction volume between the epikarst, porous media, and stream varies under different precipitation intensities. As shown in Fig. 5a, at a precipitation intensity b=3, the contribution ratios of the epikarst, PM  $\,$  I , and PM  $\,$  II to the recharge of the stream are similar. This indicates that during the early stage of precipitation, the recharge effects

of each medium on the stream are relatively balanced. Since groundwater vertically recharges the underlying aquifer through the epikarst, the discharge peak of PM  $\,$  II  $\,$  is relatively delayed compared to the epikarst and PM  $\,$  I  $\,$ 

As the precipitation intensity increases (b = 5), the contribution ratios of the epikarst, PM I, and PM II to the recharge of stream experience significant changes (Fig. 5b). Upon comparing Fig. 5a and 4b, it is evident that an increase in precipitation intensity leads to higher discharge volumes for both PM I and PM II, with PM II experiencing a more pronounced rise. Additionally, the peaks of their discharges occur earlier. The first peak of PM I is primarily caused by infiltration recharge from precipitation. With the increase in precipitation intensity, the infiltration velocity accelerates and the recharge volume increases, leading to a larger discharge volume and an earlier peak for PM I (vertical recharge peak). Groundwater continues to move downward from PM I, and the saturation of PM II rises, allowing more groundwater to overflow and discharge through PM II, thereby generating the second peak (overflow peak). For PM II, as discussed in Section 3.1, increase in saturation reduces the recharge from stream, but the discharge volume increases gradually after the middle stage of precipitation, and its contribution to the recharge of the stream becomes dominant among the three. This is due to the increased precipitation intensity, which allows PM II to receive more vertical recharge, enhancing its discharge capacity. When the precipitation intensity continues to increase (b = 7, Fig. 5c), PM II gradually reaches saturation. According to the analyses in Section 3.1, the ability of PM II to receive recharge is limited by its own saturation level, making it difficult to receive vertical recharge. Therefore, despite the increased precipitation intensity, the discharge volume of PM II does not increase significantly. Conversely, due to the influence of the saturation state of the underlying aquifer medium, the second peak (overflow peak) of PM I is more pronounced, indicating a more evident overflow phenomenon. Under higher precipitation intensity, the recharge contribution of PM I to the stream dominates. Thus, variations in precipitation intensity notably influence the interaction volume between

the karst media and stream. As precipitation intensity increases, the discharge volume and peak values of each medium are altered. Specifically, the two peaks of PM I show sequential changes in intensity, which are modulated by the saturation levels of the adjacent media.

# 3.1.3 Interaction process between the karst aquifer and stream during early stage of precipitation

The DBS model, leveraging its fine grid resolution and two-phase flow simulation capability, can accurately capture the interactive processes between various media (e.g., saturated-unsaturated zones, conduit-stream systems) influenced by dynamic saturation processes during precipitation infiltration recharge. As the interactions between adjacent media are governed by variations in saturation levels, the numerical results under rainfall intensity b=5 are selected for further analysis of dynamic inter-media interactions. For instance: How does the threshold attainment of storage capacity in the lower porous media affect the hydrological processes of the upper porous media?

As shown in Fig. 6, the Darcy-Brinkman-Stokes model clearly demonstrates the changes in the saturation levels of epikarst, porous media, and the karst spring; the saturation fields and the interaction between various media at 4000 s, 6105 s, and 7363 s; the interaction amounts between epikarst, porous media I, II, III, and the stream. From Fig. 6 (a.1), it can be seen that the saturation level of epikarst rises and declines earliest, but the saturation level is relatively low, and it is in a completely unsaturated flow state. Porous media I and III rise synchronously before 5000 s, while porous media II and the karst spring rise rapidly at 4611 s. At 7409 s, the karst spring and porous media I successively enter the decline stage. Due to the rapid drainage of the conduit, the saturation level decreases. The saturation level of the karst spring decreases faster than that of porous media I and intersects with porous media I at 9670 s.

Combining Fig. 6 (a.2) with other sub-figures, the stages with obvious interactions among porous media can be divided into the infiltration stage (green), the overflow stage (red), and the recession stage (blue). During the infiltration stage from 4000 s to 4611 s, as shown in Fig. 6 (a.2.1), epikarst vertically replenishes porous medium I and infiltrates downward. However, the infiltrating water does not reach the lower media. Meanwhile, the saturation levels of porous media II, III, and the conduit gradually increase (see Fig. 6 (a.1)). Combining with Fig. 6 (a.3), it can be seen that epikarst laterally replenishes the stream, and quickly drops to the bottom of the riverbed due to gravity. At this time, the lower aquifer system (porous media II, III, and the conduit) is in a dry state, so the stream replenishes the lower aquifer. The amount of recharge received by

porous medium III and the conduit is less than that of porous medium II (analyzed by combining Fig. 6 (a.3) and (a.4)), but their saturation levels increase faster. There are two reasons for this situation: First, the bottom elevation of the conduit is 1 m, and the water level of the stream needs to submerge the 1 m water level before it can recharge the conduit. Second, porous medium III is not only replenished by the stream, but also the sinkhole diverts the groundwater in epikarst and porous medium I to the conduit (the sinkhole flow velocity and saturation as shown in Fig. 6 (a.2.1)), and then replenishes porous medium III. As the lower aquifer media gradually tends to be saturated with rainfall recharge, as shown in Fig. 6 (a.2.2), porous media II and III tend to be saturated (see Fig. 6 (a.2.1)). Due to the weak compressibility of water, after the upper part infiltrates and replenishes porous medium I, it tends to laterally replenish the stream from the interface between porous medium II and stream. As the saturation level of porous medium I gets higher, the lateral recharge to the stream becomes more significant, showing an obvious overflow state. The depression between the two peaks is caused by the rapid rise of the stream water level. During the flood peak stage, the discharge from porous media to stream decreases. At the same time, the rise of the stream water level makes it difficult for the lower porous media to replenish the stream, and porous medium II tends to be saturated, making it difficult to replenish porous medium I. During this stage, the flow between porous media I and II is in a dynamic equilibrium state. As shown in Fig. 6 (a.2.3), during the recession stage, the rainfall infiltration intensity decreases rapidly. Under the action of gravity, the groundwater vertically replenishes porous medium I, the conduit, and porous medium II successively recedes. And the water level of the stream drops rapidly (see Fig. 3 (e)). The groundwater tends to be discharged to the stream through porous medium I and the karst spring. Porous medium I is replenished by porous medium II on the one hand and discharges to the stream on the other hand. Therefore, during a single rainfall event, during the infiltration stage, part of the amount of water replenished from epikarst to the stream is discharged, and other part is redirected to replenish the lower porous media; during the overflow stage, the stream is mainly replenished through the karst conduit and porous medium II. Porous medium I and the stream are in a dynamic equilibrium state. During the recession stage, the porous media act as the main medium to replenish the stream.

As shown in Fig. 6 (a.4), the karst spring reaches its peak at 7409 s. This is due to the rainfall infiltration, the recharge from porous medium I, and the subsequent discharge to the stream. As the storage volume decreases, the amount of recharge from the karst spring to the stream decreases. A trough appears at 11642 s. This is because as the water level of the stream drops, groundwater is more easily discharged into the stream. However, as the overall storage volume continues to decline, after a peak appears at 13057 s, it enters a complete recession stage. Affected by the decline of the stream water level, the discharge from porous medium III to the stream gradually increases during the recession stage. Combining with Fig. 6 (a.1), it can be seen that while porous medium III is discharging, its saturation remains at level I continuously, indicating that the conduit continuously supplies water vertically to porous medium III.

Under the recharge of rainfall infiltration, the interaction process between the karst aquifer affected by epikarst, sinkholes, conduit and the stream shows dynamic changes in terms of staged characteristics, main interaction media, and the dynamic equilibrium process among different media. The accurate simulation of the above complex processes depends on the support of a three-dimensional two-phase numerical model (Darcy-Brinkman-Stokes model).

# 3.2. Impact of multiple precipitation events on the interaction process between the karst aquifer and stream

Rainy seasons typically experience multiple precipitation events, during which differences in precipitation peaks, durations, and cumulative precipitation events can all impact the interaction process between the karst aquifer and stream. Does the groundwater stored in the porous media of the karst aquifer system during the initial rainfall event influence the interactions between multi-component media during subsequent precipitation episodes?

Based on understanding the interaction mechanism of a single precipitation event, this study further analyzes the impact of multiple precipitation events on the interaction process. Figure 7 shows the changes in water level of stream under continuous precipitation events. When the intensities of two consecutive precipitation events remain constant, the water level of stream reaches both the highest and the lowest points, indicating that the water level is related to the total precipitation intensity. Even with different intensities of the first precipitation event ( $b_1 = 3$  and  $b_1$ 

=5), the trend of the water level changes in stream is consistent (Fig. 70 and 4). After the first precipitation event, the karst aquifer receives infiltration recharge from the precipitation and can store part of the water, so the water level of stream will be higher during the second precipitation event, and the greater the intensity of the second precipitation event, the higher the water level of stream (Fig. 71) and 2, or 3 and 4). This indicates that the intensity of the second precipitation event determines the amount of recharge from each medium to stream. Therefore, when the intensity of the first precipitation event is the same, the amplitude of the water level change in stream during the second precipitation event is only related to the intensity of the second precipitation event. When the intensity of the second precipitation event is the same, the storage capacity of the karst aquifer during the first precipitation event determines the amplitude of the water level change in stream during the second precipitation event. When the total precipitation intensity is the same (Fig.  $7 \otimes 2$  and 3), if the intensity of the first precipitation event is lower than that of the second one, the amplitude of the water level change in stream is higher, and vice versa. This is because, in the case of two consecutive precipitation events, part of the precipitation infiltrates and recharge the storage during the first event, and the other part is discharged to stream through the aquifer. Combining Fig. 4d and e, during the first precipitation event, the water level in the porous medium rises and stores a proportion of water, but the discharge volume to stream is greater when the precipitation intensity is higher ( $b_1=5$ ) compared to when it is lower ( $b_1 = 3$ , Fig. 4a and b). When the second precipitation event occurs, due to the similar saturation levels of the karst aquifer, the greater the intensity of the second precipitation event, the larger the amount of groundwater recharged to stream through the aquifer, and the more pronounced the amplitude of the water level in stream. Figure 8 illustrates the hydrological process curves of the stream during two consecutive

Figure 8 illustrates the hydrological process curves of the stream during two consecutive precipitation events, as well as the interaction processes between the various media of the karst aquifer and stream. Under different precipitation intensities, the various media of the karst aquifer recharge the stream with varying intensities, resulting in significant fluctuations in the water level of stream. Based on Fig. 8(a) and Fig. 7 (2 and 4), it can be observed that under two consecutive precipitation events, when the intensity of the second precipitation event is equal to or greater than

the first, the stream hydrograph exhibits more pronounced fluctuations. The comparison between the DBS model and MODFLOW-CFPv2 model under different b<sub>1</sub> parameter combinations demonstrates distinct characteristics in streamflow hydrographs: the DBS model shows higher peak discharge with greater fluctuations, while the MODFLOW-CFPv2 model displays relatively smoother discharge variations. Notably, under the second precipitation event, the MODFLOW-CFPv2 model exhibits delayed peak elevation timing. Furthermore, its recession phase still follows an exponential decay pattern, failing to capture the rapid interactive response between multimedia systems during successive precipitation events. As shown in Fig. 8b, the epikarst discharges quickly and is not easily affected by multiple precipitation events. However, when the intensity of the first precipitation is high and the intensity of the second precipitation is the same (1) and (3), the discharge volume of the epikarst to stream during the second precipitation period is slightly larger. When the intensity of the first precipitation is different and the intensity of the second precipitation is the same (Fig. 8c 2) and 4), the discharge volume of groundwater through karst conduit to stream during the second precipitation period is almost the same. This is because karst conduit discharge quickly, and the storage volume of the conduit during the first precipitation period has little impact on the storage volume during the second precipitation period. Therefore, combining with Fig. 7, it is known that the storage effect of the karst aquifer mainly occurs in the porous medium, and it also indicates that relying solely on changes in the water level of stream makes it difficult to clearly determine the storage volume of the porous medium and conduit during the first precipitation event, and their respective impacts on the second precipitation period (Fig. 7). When the intensity of the second precipitation is higher (Fig. 8c ②, ③ and ④), the discharge volume of the porous medium (PM II) to stream does not increase significantly. This is because the intensity of the second precipitation is larger, causing the water level of stream to rise (Fig. 7), making it difficult for the porous medium (PM II) to recharge stream.

Therefore, under the influence of two consecutive precipitation events, the greater the total precipitation intensity, the larger the discharge volume of the karst aquifer to stream. The storage effect of the karst aquifer occurs in the porous medium and affects subsequent precipitation processes. The lower-level porous medium (PM II), due to the high water level and large

fluctuations of stream, is more difficult to recharge stream, and the recharge from stream mostly comes from overflow supply from the media in other layers.

### 3.3. Effects of Water Retention Characteristics on Karst Aquifer-Stream Interactions

The external recharge of the system significantly influences the interaction processes among different media. This study further investigates how the inherent hydrogeological properties of karst systems affect these interactive processes. Variable saturated flow in the karst vadose zone plays a critical role (Dvory et al., 2018), where the water retention characteristics of porous media govern unsaturated flow dynamics. However, the CFPv2 model struggles to simulate variable saturation processes. This paper compares the DBS model results with two distinct experimental datasets to elucidate the advantages and limitations of the DBS approach in simulating variable saturated flow.

Case 1: A typical unsaturated-unsteady seepage problem in sandy clay loam (Warrick et al., 1985), where the soil hydraulic properties are provided by the international UNSODA database (Leij et al., 1996). Key parameters include:  $k = 1 \times 10$ –6 m/s,  $\alpha_s = 0.363$ ,  $\alpha_r = 0.186$ , and n = 1.53. The model consists of a vertical soil column (1 m thickness) with an initial pressure head of -8 m across the domain. The top boundary is set to a pressure head of 0 m to simulate free surface infiltration.

Case 2: A 2D laboratory infiltration experiment by Vauclin et al. (1979), widely used for evaluating saturated-unsaturated unsteady seepage models. The soil slab measures 2.00 m in height, 6.00 m in width, and 0.05 m in thickness, with an impermeable base and free drainage boundaries on both sides. Initially, the water table is set at 0.65 m. A central 1.00 m section of the top boundary receives uniform precipitation at 0.148 m/h for 8 hours, during which free surface evolution is monitored. Soil hydraulic properties are described using the van Genuchten-Mualem model with parameters:  $k = 0.35 \, m/h$ ,  $\alpha_s = 0.30$ ,  $\alpha_r = 0.01$ . Due to symmetry, the DBS model simulates the right half of the domain.

As shown in Fig. 9, the DBS model demonstrates strong agreement with both experimental datasets, highlighting its capability to capture spatiotemporal variations in water-air two-phase flow. Comparative analysis between DBS simulations and experimental data not only validates model reliability but also enhances understanding of soil moisture transport mechanisms. This provides critical support for simulating interactions between karst aquifers and adjacent streams.

Based on the well-validated two-phase flow DBS model, this study analyzes the impacts of different

water retention models on interactive flow between media. Fig. 10 presents the hydrograph curves under different water retention model parameters (BCn=3, 2.5, 2 and VGMm=0.85, 0.8) for (a) stream, (b) karst spring, (c) epikarst, (d) PM I, (e) PM II, and (f) PM III. Fig. 10(c.1) illustrates the parameter effects on porous media morphology, where n≥2 and higher n values indicate more heterogeneous pore space and complex structures. Fig. 10(d.1) compares water retention curves between BC and VGM models.

Combining Figs. 10(a) and (b), in the BC model, increasing n values progressively reduce hydrograph curves of stream and karst spring, attributed to irregular pore media impeding groundwater flow and reducing discharge. In the VGM model, decreasing m values (equivalent to increasing n) enhance pore structure irregularity, similarly lowering hydrograph curves. As shown in Fig. 10(c), epikarst discharge increases with higher n values due to its low permeability (K0) during relative permeability correction, facilitating enhanced groundwater discharge through epikarst to the stream.

From Figs. 10(d) and (e), larger n values correspond to decreased epikarst-stream discharge and increased downward recharge to porous media, thereby enhancing stream recharge from PM I and II. Integrating Figs. 10(c) and (e), reduced epikarst-stream hydrographs with higher n values lead to diminished stream-porous media recharge. Fig. 10(f) demonstrates that PM III is primarily influenced by conduit flow and shows minimal sensitivity to n and m parameters.

Fig. 10(d.1) displays saturation variations derived from two karst groundwater retention models: Brooks-Corey (BC) model (Equations (20)-(21)) and van Genuchten-Mualem (VGM) model (Equations (22)-(23)). For identical infiltration periods, BC model predicts higher moisture retention than VGM. The BC model emphasizes static water retention in karst media, while VGM prioritizes dynamic groundwater transport and distribution. The VGM model predicts longer groundwater migration distances, suggesting greater sensitivity in simulating karst groundwater diffusion and infiltration processes. These differences hold significance for unsaturated two-phase flow dynamics and accurate prediction of groundwater migration paths in karst aquifer systems.

Furthermore, discrepancies exist between BC and VGM models in simulating saturation variations (Fig. 10(d.1)), manifesting as distinct saturation degrees and groundwater migration distances under identical conditions. Therefore, selecting appropriate models based on lithological characteristics is crucial for precise description and prediction of two-phase flow in karst groundwater systems.

# 3.4. Impact of multi-stage permeability and porosity arrangement on the interaction process between the karst aquifer and stream

By comparing the effects of multi-level versus single-level conduit configurations on interactive processes, the adoption of both multi-level and single-level conduits in the karst conduit system and underlying media did not induce significant changes in the hydrological processes of the epikarst and porous media (I, II) (Fig. 11). As shown in Fig. 11a, when multi-level conduit arrangements are adopted, the peak of stream hydrological process increases, indicating that multi-level conduit arrangements enhance the recharge volume of stream. However, during the recession phase, the flow under multi-level conduit arrangements is relatively low. This is because multi-level conduit collects a proportion of the flow that should have been contributed by the later stage matrix recession and discharge it to stream, thereby affecting the peak of the recession process. As shown in Fig. 11b, under multi-level conduit arrangements, sinkhole can absorb more water and discharge it through karst conduit. This indicates that multi-level conduit arrangements can more effectively play their roles in water absorption and discharge during heavy precipitation events. However, in the case of lower precipitation intensity in the early stage, the water absorption priority of multi-level conduit is not fully manifested. By comparing Figs. 11c, 11d, and 11e, it is found that multi-level conduit arrangements have no significant impact on the hydrological processes of the epikarst and porous media (PM I and PM II). This suggests that multi-level conduit arrangements mainly affect the interaction between the karst conduit and stream, with relatively little impact on other media. The hydrological responses of the karst conduit and PM II under multi-level conduit arrangements are shown in Figs. 11f and 11b. Under multi-level conduit arrangements, the discharge volume of the karst conduit significantly increases. At the same time, due to the increase in karst conduit flow, PM II also receives more recharge, leading to a corresponding increase in the discharge volume of this portion of porous media to stream. This further indicates that multi-level conduit configurations can notably influence the hydrological processes of stream and karst conduit under specific precipitation intensities, with minimal effects on other media.

# 4 Uncertainty Analysis and Discussion

The multi-level conduit configuration inherently affects multi-media interactions by simultaneously

altering permeability, conduit diameter, and porosity parameters. This study will further conduct sensitivity analyses on individual variables to investigate their impacts on the vulnerability of karst aquifer systems.

# 4.1 Impacts of Conduit Diameter and Geometry on Interactions Between Karst Aquifer Systems and Streams

Fig. 12 presents hydrographs under conditions of circular conduits with varying radii (r=0.2, 0.3, 0.3, and 0.5 m) and square-section conduits (r=0.5 m) for (a) stream-connected flow, (b) karst spring discharge, (c) epikarst flow, (d) porous medium I (PM I), (e) PM II, and (f) PM III. Fig. 12(c.1) illustrates different conduit cross-sectional shapes to analyze their impacts on the interactive flow between karst aquifer systems and adjacent streams.

As shown in Fig. 12(a), larger conduit radii correspond to higher initial discharge peaks and shorter peak arrival times, indicating enhanced porous medium recharge and faster fluid transmission through larger conduits. Notably, the square-section conduit (s-r<sub>c</sub>=0.5) exhibits higher peak discharge than its circular counterpart (rc=0.5) due to its surplus cross-sectional area accommodating greater fluid discharge under identical nominal radii.

Fig. 12(b) demonstrates that karst spring peak discharge increases with conduit radius. At r=0.5 m, the square-section conduit ( $s-r_c=0.5$ ) achieves higher peak discharge than the circular conduit ( $r_c=0.5$ ), but displays lower recession flow. This occurs because identical precipitation infiltration recharge leads to greater porous medium storage depletion during peak periods in square conduits, subsequently reducing porous medium-to-conduit recharge during baseflow recession.

Combined analysis of Figs. 12(c), (d), and (e) reveals that conduit radius variations do not significantly affect epikarst hydrographs or PM I/II hydrographs. However, square-section sinkholes modify flow patterns: epikarst hydrographs show lower values under square conduits, while PM I/II hydrographs exhibit higher values due to enhanced epikarst groundwater collection in square cross-sections, increasing recharge to PM I/II.

Fig. 12(e) indicates that larger conduit radii correspond to lower negative values. Combined with Fig. 12(a), this demonstrates that increased stream recharge through larger conduits elevates both stream peak discharge and water levels, thereby enhancing porous medium-stream interactions. Similarly, Fig.

12(f) shows that larger conduit radii increase karst spring discharge and PM III hydrograph elevation through enhanced gravity-driven groundwater recharge.

Conduit geometry (radius and shape) constitutes a critical factor in karst aquifer hydrological modeling. Larger circular conduits accelerate peak discharge arrival and amplify stream-connected flow peaks and karst spring discharge. Square-section conduits outperform circular equivalents in peak discharge capacity under identical nominal radii due to cross-sectional area advantages. Enlarged conduits intensify porous medium-stream interactions and amplify PM III recharge through gravitational effects. Comprehensive consideration of conduit geometry impacts on hydrological elements is essential for improving model accuracy and reliability in simulating karst aquifer-stream interaction processes.

# 4.2 Influence of Permeability on the Interaction Processes Between Karst Aquifer Systems and Streams

The permeability of the epikarst directly controls the ease of fluid infiltration from the surface into the conduit system. Fig. 13 illustrates the hydrological process curves under different epikarst permeability coefficients ( $K_E=10^{-6}$ ,  $10^{-7}$ ,  $10^{-8}$ ,  $10^{-9}$ ; when  $K_E=10^{-9}$ , the permeability matches that of porous media, rendering the epikarst incapable of rapid groundwater leakage) for: (a) stream, (b) karst spring, (c) epikarst, (d) PM I, (e) PM II, and (f) PM III. This aims to reveal how epikarst permeability regulates groundwater flow patterns in complex conduit systems and intermedia interactions.

As shown in Fig. 13(a), under high epikarst permeability ( $K_E=10^{-6}$ ): the discharge curve rises rapidly to a peak of ~4.5  $m^3/s$  followed by a sharp decline. This indicates that high permeability enables rapid groundwater leakage from the epikarst to the stream, causing swift flow increases. Peak stream discharge diminishes with decreasing permeability. High permeability reduces flow resistance, facilitating faster fluid entry into the conduit system and generating sharp discharge peaks, while low permeability increases resistance, resulting in gradual fluid release and broader, lower discharge curves.

Fig. 13(b) demonstrates that epikarst permeability differences from porous media have minimal impact on conduit flow. However, when epikarst permeability equals that of porous media ( $K_E=10^{-9}$ ), the peak discharge at the karst spring decreases while maintaining identical baseflow recession characteristics. Combining Figs. 13(c) and (c.1), higher epikarst permeability enhances lateral discharge to the stream. At  $K_E=10^{-9}$ , gravitational forces dominate vertical recharge to lower media without lateral

discharge.

Fig. 13(d) reveals decreasing discharge from Porous Medium I to the stream with reduced epikarst permeability. Cross-referencing Figs. 13(a) and (e), lower epikarst permeability reduces both stream discharge and water level, limiting recharge to Porous Medium II. Fig. 13(f) shows negligible epikarst permeability influence on Porous Medium III's hydrograph.

Epikarst permeability constitutes a critical factor in hydrological modeling of karst aquifer systems. Highly permeable epikarst produces rapid streamflow peaks followed by sharp declines, reflecting efficient groundwater leakage to the stream. Conversely, low permeability yields diminished peaks and broader discharge curves. While karst spring discharge remains relatively stable when epikarst permeability differs from porous media, proper characterization of epikarst permeability is essential for accurately simulating hydraulic interactions between media, regulating groundwater flow pathways and velocities. This enhances model reliability in capturing complex flow dynamics within karst conduit-stream systems.

# 4.3 Influence of Porosity on the Interaction Between Karst Aquifer Systems and Adjacent Streams

Fig. 14 presents the hydrographic process curves under different porosity conditions ( $\varphi$ =0.4,  $\varphi$  =0.3,  $\varphi$  =0.2,  $\varphi$  =0.1) for (a) stream, (b) karst spring, (c) epikarst, (d) PM I, (e) PM II, and (f) PM III. Fig. 14(c.1) illustrates the schematic diagram of groundwater flow under different pore sizes. The study aims to elucidate how porosity regulates fluid flow patterns in complex conduit systems.

As shown in Fig. 14(a), lower porosity results in higher flow peaks and earlier peak times. This occurs because reduced pore space limits groundwater storage capacity, forcing excess water to discharge rapidly and elevating the stream hydrograph. Fig. 14(b) demonstrates that lower porosity drives groundwater to preferentially flow through karst conduits and discharge at springs. In Fig. 14(c), the peak discharge of epikarst at  $\varphi = 0.4$  slightly exceeds those at  $\varphi = 0.3$ ,  $\varphi = 0.2$ , and  $\varphi = 0.1$ .

Fig. 14(d) reveals that at  $\varphi$  =0.1, the storage capacity of porous medium I reaches critical limits. Groundwater recharged from epikarst to porous medium I is rapidly discharged, resulting in significantly higher discharge rates compared to  $\varphi$  =0.3,  $\varphi$  =0.2, and  $\varphi$  =0.1. Fig. 14(e) indicates increased discharge from porous media to the stream as porosity decreases. Combined with Fig. 14(a), reduced porosity enhances stream stage and discharge but diminishes the stream's ability to recharge porous media due to

limited storage capacity. Fig. 14(f) shows negligible porosity effects on the hydrograph of porous medium III, as its behavior is primarily governed by conduit flow.

In hydrological modeling, porosity parameters must be calibrated to accurately simulate groundwater flow paths and storage-release dynamics. For low-porosity regions, models should emphasize rapid drainage capacity of conduit systems and transient flow variations. In high-porosity areas, considerations should include fluid retention risks, stream-porous media interactions, and their long-term impacts on geological stability and water resource allocation. Proper porosity parameterization enhances simulation accuracy for diverse hydrological processes, enabling improved prediction and management of karst water resources.

Karst hydrological vulnerability manifests prominently through rapid infiltration, epikarst runoff, groundwater table fluctuations, and abrupt spring discharge variations. The DBS model effectively simulates multi-media interactions during extreme recharge events, enabling temporal analysis of mediastream exchanges, identification of peak interaction values, and applications in coupled conduit flow-seepage processes for two-phase flow systems.

# **5 Conclusions**

This study employed the Darcy-Brinkman-Stokes equation to characterize groundwater flow in the karst aquifer and stream, as well as within the karst media. The VOF phase change method was used to illustrate the two-phase flow of water and air in porous media, while various water retention models were applied to describe the unsaturated flow processes in the karst aquifer. The results indicate that changes in precipitation intensity have a significant impact on the interaction between the karst aquifer and stream. As the precipitation intensity increases, the interaction process between the two becomes more complex, involving multi-media synergistic recharge and dynamic interaction with the karst aquifer. The contribution ratios of the epikarst, upper layer, and PM II to the stream change with increasing precipitation intensity. In the early stages of precipitation, the recharge effects of each medium on the stream are relatively balanced; as the precipitation intensity increases, the discharge volumes of PM I and PM II both increase, especially the increase in PM II is more significant, and the timing of its discharge peak advances; when the precipitation intensity further increases, PM II gradually reaches

saturation, limiting its discharge capacity; and during this process, the double peak intensity of PM I changes with the precipitation intensity; at the same time, due to the saturation of PM II, a more pronounced overflow phenomenon occurs in PM I, which dominates the contribution of recharge volume to the stream. Therefore, the change in precipitation intensity not only affects the discharge volume and discharge peak of each medium in the karst aquifer but also is influenced by the dynamic saturation process of adjacent media. By analyzing the modeling differences between MODFLOW-CFPv2 and DBS for the conceptualized model of this study and conducting comparative validation through stream hydrographs, results demonstrate that the DBS model can effectively simulate the interaction process between karst aquifer systems and adjacent streams under precipitation influences, while refining two-phase interactive flows between different media subjected to dynamic saturation processes.

Under two consecutive precipitation events, total rainfall intensity directly governs stream water level variations. Different rainfall intensities induce distinct changing trends in stream water levels. During the first rainfall period, porous media in the karst aquifer system store a portion of groundwater, which subsequently influences stream water level changes in the second rainfall period. Due to the rapid drainage characteristics of karst conduits, the storage capacity of conduits during the first rainfall period shows negligible impact on storage during the second rainfall period. When the first rainfall intensity exceeds the second, stream water level fluctuations exhibit smaller amplitudes, and vice versa. Variations in stream water levels can alter the recharge potential from different layered media in the karst aquifer system to the stream. Different water retention models also demonstrate significant impacts on hydrological processes in both the stream and various media. The accuracy of two-phase flow simulation in the DBS model was validated against benchmark experiments from two literature sources. The VGM model causes greater water retention in porous media, thereby reducing stream discharge.

During heavy rainfall events, multi-level conduit configurations significantly affect interaction processes between karst aquifer systems and adjacent streams, demonstrating higher drainage efficiency. However, such configurations exhibit relatively minor impacts on other media, indicating that multi-level conduit arrangements primarily influence hydrological processes by regulating interactions between karst conduits and the stream.

In uncertainty analysis: For circular conduits, larger diameters result in higher initial peak discharge

in streams and shorter time-to-peak, with corresponding increases in peak discharge from karst springs. Under identical diameters, square-section conduits demonstrate higher peak stream discharge and karst spring discharge than circular counterparts due to surplus space advantages. Epikarst permeability significantly influences hydrological processes in karst aquifer systems. High-permeability epikarst produces rapid stream discharge peaks followed by steep recessions. With decreasing permeability, peak stream discharge diminishes and hydrographs become lower and broader. Concurrently, karst spring peak discharge decreases, with epikarst only vertically recharging underlying media without lateral discharge. Reduced epikarst permeability decreases discharge from porous media to streams.

Porosity proves crucial in governing hydrological processes of karst aquifer systems: Lower porosity leads to higher and earlier discharge peaks in both streams and karst springs, as reduced pore spaces limit groundwater storage and force faster drainage. Higher porosity results in lower peaks and broader hydrographs. Decreasing porosity increases discharge from porous media to streams but reduces the stream's recharge capacity to porous media due to diminished storage space. Hydrological modeling should prioritize rapid drainage and transient flow variations in conduit systems for low-porosity areas, while high-porosity regions require consideration of fluid retention risks, interactive flows between streams and porous media, along with long-term impacts on geological stability and water resource allocation.

#### Acknowledgments

- This research was partially funded by the Doctoral Scientific Research Startup Foundation of Xinjiang
- University grant 620321004, and the Natural Science Foundation of Xinjiang Uygur Autonomous Region
- grant 2022D01C40.
- Data availability. All raw data can be provided by the corresponding author upon request.
- Author contributions. FH: conceptualization, methodology, formal analysis, visualization, writing
- original draft. YG: conceptualization, methodology, formal analysis, visualization, review and editing.

- **ZZ:** methodology, formal analysis, visualization, review and editing. **XH**: visualization, review and
- editing. XW: methodology, review and editing. SP: writing original draft and review and editing.
- **Competing interests.** The authors declare that they have no conflict of interest.

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

Table 1: Variable Definition Table

| Variable                            | Description                                               | Unit         |  |
|-------------------------------------|-----------------------------------------------------------|--------------|--|
| φ                                   | Porosity field                                            |              |  |
| V                                   | Volume of the averaging-volume                            | $m^3$        |  |
| $V_l$                               | Water Volume                                              | $m^3$        |  |
| $V_g$                               | Gas Volume                                                | $m^3$        |  |
| $\alpha_l$                          | Water Saturation                                          |              |  |
| $lpha_g$                            | Gas Saturation                                            |              |  |
| $\alpha_{l,\mathrm{e}}$             | Effective Saturation                                      |              |  |
| ρ                                   | Average Fluid Density                                     | $kg/m^3$     |  |
| $ ho_g$                             | Gas Density                                               | $kg/m^3$     |  |
| $ ho_l$                             | Water Density                                             | $kg/m^3$     |  |
| μ                                   | Viscosity                                                 | Pa·s         |  |
| $\mu_g$                             | Gas Viscosity                                             | $Pa \cdot s$ |  |
| $\mu_l$                             | Water Viscosity                                           | $Pa \cdot s$ |  |
| $\mu_{eff}$                         | effective viscosity                                       | $Pa \cdot s$ |  |
| $\frac{\overline{v}}{\overline{v}}$ | velocity                                                  | m/s          |  |
| $\overline{v_r}$                    | relative flow rate of the gas phase to the liquid phase   | m/s          |  |
| $v_t^{'}$                           | turbulent velocity vector                                 | m/s          |  |
| $v_{rt}$                            | relative velocity of gas-phase and water-phase turbulence | m/s          |  |
| $oldsymbol{v}_{turb}$               | turbulent kinetic viscosity                               | $m^2/s$      |  |
| $\bar{p}$                           | pressure                                                  | Pa           |  |
| $p^*$                               | pressure                                                  | Pa           |  |
| $F_c$                               | Surface tension force                                     | N            |  |
| $S_f$                               | Drag Source Term                                          | $N/m^3$      |  |
| $C_{\mu}$                           | Dimensionless Constant                                    |              |  |
| $k_t$                               | Turbulent Kinetic Energy                                  | $m^2/s^2$    |  |
| ε                                   | Turbulent Dissipation                                     | $m^2/s^3$    |  |
| k                                   | Apparent permeability                                     | $m^2$        |  |
| $k_0$                               | Absolute permeability                                     | $m^2$        |  |
| $k_{rg}$                            | Gas Relative Permeability                                 |              |  |
| $k_{rl}$                            | Water Relative Permeability                               |              |  |
| g                                   | Gravitational Acceleration                                | $m/s^2$      |  |
| X                                   | position vectors in Cartesian                             | ,            |  |
| $\sigma$                            | Interfacial tension                                       | N/m          |  |
| $p_c$                               | Capillary pressure                                        | Pa           |  |
| n                                   | Brooks and Corey Coefficient                              |              |  |
| m                                   | Van Genuchten Coefficient                                 |              |  |

Table 2: Different parameter used in Models

| Parameters                                   | Unit     | Value     |
|----------------------------------------------|----------|-----------|
| Conduit radius $r_c$                         | m        | 0.5       |
| Sinkhole radius $r_s$                        | m        | 0.5       |
| Conduit height $h_S$                         | m        | 2         |
| River width $L_r$                            | m        | 2         |
| EpiKarst thickness                           | m        | 4         |
| Porous medium I thickness                    | m        | 13        |
| Porous medium II thickness                   | m        | 3         |
| Porous medium III thickness                  | m        | 1         |
| Porous medium length $L_{py}$                | m        | 200       |
| Porous media width $L_{px}$                  | m        | 200       |
| Gravitational acceleration $g$               | $m/s^2$  | 9.81      |
| Porous medium Porosity $\varphi$             | /        | 0.4       |
| Porous medium Permeability coefficient $k_0$ | $m^2$    | $10^{-9}$ |
| Gas phase viscosity $\mu_a$                  | $m^2/s$  | 1.48*10-5 |
| Gas phase density $\rho_a$                   | $Kg/m^3$ | 1.29      |
| Liquid phase viscosity $\mu_w$               | $m^2/s$  | 10-6      |

Table 3: Comparing DBS and MODFLOW results for key variables

| Numerical<br>Model | Peak Lag Time (s) |             | Peak Flow $(m^3/s)$ |       | Total Outflow (m <sup>3</sup> ) |              |              |               |               |
|--------------------|-------------------|-------------|---------------------|-------|---------------------------------|--------------|--------------|---------------|---------------|
|                    | b=3               | b = 5       | <b>b</b> = 7        | b = 3 | b = 5                           | <b>b</b> = 7 | b=3          | b = 5         | <b>b</b> = 7  |
| DBS Model          | 3242.<br>96       | 1870.<br>18 | 2985.<br>31         | 4.50  | 12.14                           | 21.96        | 65984<br>.49 | 15415<br>8.46 | 27294<br>5.87 |
| MODFLOW<br>-CFPv2  | 2520.<br>00       | 1920.<br>00 | 1860.<br>00         | 4.31  | 11.87                           | 18.87        | 63916        | 15754<br>3.65 | 24551<br>9.26 |

Figure 1. Schematic diagrams of the modelling of the interaction between the karst aquifer (epikarst, sinkhole, karst conduit, PM  $\parallel$ , PM  $\parallel$ , and PM  $\parallel$ ) and stream under dimensionless precipitation intensities (b=3 and b=5). (a) and (a.1) Schematic diagram of the interaction flow between each medium and stream in the early stage of a precipitation event; (b) and (b.1) Schematic diagram of the interaction flow between each medium and stream in the middle stage of a precipitation event. The size of the arrows represents the magnitude of the flow rate, and the direction of the arrows represents the direction of interaction between the two.

Figure 2. Diagram of performance and applicability of different models, (a) N-S model (Navier-Stokes model), (b) DBS model, (c) Schematic diagram of MODFLOW-CFP model solution, (d) Conversion method from DBS equations to N-S equations and Darcy equations.

Figure 3. (a) Schematic comparison of conduit and porous media coupling modes between MODFLOW-CFPv2 and DBS, (b) DBS model and (c) CFPv2 discretization schemes for karst aquifer systems with riverside models.

Figure 4. Hydrological process curves of each medium in the karst aquifer and stream for different precipitation intensities: (a) b = 3, (b) b = 5, (c) b = 7. Water level changes and differences in water levels in the karst aquifer and stream for different precipitation intensities: (d) b = 3, (e) b = 5, (f) b = 7.

Figure 5. Interaction process of epikarst, porous media, and stream for different precipitation intensities: (a) b = 3, (b) b = 5, (c) b = 7.

Figure 6. For the Darcy-Brinkman-Stokes model: (a.1) Variations in the saturation levels of epikarst, various porous media, and the karst spring. (a.2) Saturation fields and the interaction among different media at 4000 s, 6105 s, and 7363 s. (a.3) Interaction volumes between epikarst, porous media I, II, and the stream. (a.4) Interaction volumes among the karst spring, porous media III, and the stream.

Figure 7. Water levels in stream for two consecutive precipitation events with first and second precipitation intensities ①  $b_1 = 3$  and  $b_2 = 3$ ; ②  $b_1 = 3$  and  $b_2 = 5$ ; ③  $b_1 = 5$  and  $b_2 = 3$ ; ④  $b_1 = 5$  and  $b_2 = 5$ , respectively.

Figure 8. (a) Hydrological process curves of the stream; (b) Discharge process of groundwater through the epikarst to the stream; (c) Discharge process of groundwater through the karst conduit to the stream; (d) Discharge process of porous media (PM II) to the stream, for two consecutive precipitation events with first and second precipitation intensities ①  $b_1 = 3$  and  $b_2 = 3$ ; ②  $b_1 = 3$  and  $b_2 = 5$ ; ③  $b_1 = 5$  and  $b_2 = 5$ , respectively.

Figure 9. Comparison between the DBS model and experimental results from (a) Warrick et al. (1985) and (b) Vauclin et al. (1979).

Figure 10. Hydrological process curves under different water retention model parameters (BCn = 3, 2.5, 2 and VGMm = 0.85, 0.8) for (a) stream, (b) karst spring, (c) epikarst, (d) PM I, (e) PM II, and (f) PM III. Subplots (c.1) and (d.1) show the schematic diagram of parameter effects on porous media morphology and the water retention curves of the BC and VGM models, respectively.

Figure 11. Impacts of single-stage and multi-stage conduit hydrological process changes in various media of the karst aquifer for a precipitation intensity b = 5.

Figure 12. Hydrological process curves for (a) stream, (b) karst spring, (c) epikarst, (d) PM I, (e) PM II, and (f) PM III under conditions of circular conduits with radii rc = 0.2, 0.3, 0.3, and 0.5, and square-cross-section conduits with S-rc = 0.5. Subplot (c.1) shows a schematic diagram of different conduit cross-sectional shapes.

Figure 13. Hydrographs under different epikarst permeability conditions (KE=10<sup>-6</sup>, KE=10<sup>-7</sup>, KE=10<sup>-8</sup>, KE=10<sup>-9</sup>) for: (a) stream, (b) karst spring, (c) epikarst, (d) PM I, (e) PM II, (f) PM III. Subfigure (c.1) shows a schematic diagram of media interactions under varying epikarst permeability conditions.

Figure 14. hydrograph curves under different porosity conditions ( $\phi$  = 0.4,  $\phi$  = 0.3,  $\phi$  = 0.2,  $\phi$  = 0.1) for (a) stream, (b) karst spring, (c) epikarst, (d) PM I, (e) PM II, and (f) PM III. Among these, (c.1) illustrates a schematic diagram of the medium's water storage capacity and flow capacity under varying porosity conditions.