# Peer review of "Simulating Precipitation-Induced Karst-Stream"

_EGUsphere, 2025_

## Author Comment (AC1)

**Reply to Referee 1:**

*The article focuses on studying the interactions between hydrological processes involving karst aquifers, conduits, and streams, varying with precipitation and other factors that are later described in detail. The work is well-structured, the figures are of good quality, and the reading is fairly smooth.*

Dear Reviewer: Thank you very much for your comments and suggestions. Our replies are listed as follow:

(1) *The topic is, in my opinion, significant but somewhat self-contained when the discussion remains limited to numerical results without extending them to a broader analysis—namely, how this study could be useful in managing water resources in these complex aquifers or how it could be applied at a regional scale.*

Thank you for pointing out the need to strengthen the practical application relevance in the Discussion section. In response to this suggestion, we have added implications of modeling parameters for karst water resources management in the revised version, as follows:

- Lines 847-855: Discusses the effect of conduit geometry (e.g., square cross-section) on aquifer-stream interaction fluxes, emphasizing its critical role in enhancing simulation accuracy.
- Lines 886-894: Analyzes the regulatory mechanism of epikarst permeability on groundwater flow paths, illustrating the importance of parameter calibration for water resources prediction.
- Lines 917-930: Adds an analysis of the impact of porosity parameters on karst water resources allocation and geological stability, clarifying the model's potential application in flood warning.

Lines 847-855:

"Conduit geometry (radius and shape) constitutes a critical factor in karst aquifer hydrological modeling. Larger circular conduits accelerate peak discharge arrival and amplify stream-connected flow peaks and karst spring discharge. Square-section conduits outperform circular equivalents in peak discharge capacity under identical nominal radii due to cross-sectional area advantages. Enlarged conduits intensify porous medium-stream interactions and amplify PM III recharge through gravitational effects. Comprehensive consideration of conduit geometry impacts on hydrological elements is essential for improving model accuracy and reliability in simulating karst aquifer-stream interaction processes."

Lines 886-894:

"Epikarst permeability constitutes a critical factor in hydrological modeling of karst aquifer systems. Highly permeable epikarst produces rapid streamflow peaks followed by sharp declines, reflecting efficient groundwater leakage to the stream. Conversely, low permeability yields diminished peaks and broader discharge curves. While karst spring discharge remains relatively stable when epikarst permeability differs from porous media, proper characterization of epikarst permeability is essential for accurately simulating hydraulic interactions between media, regulating groundwater flow pathways and velocities. This enhances model reliability in capturing complex flow dynamics within karst conduit-stream systems."

Lines 917-930:

"In hydrological modeling, porosity parameters must be calibrated to accurately simulate groundwater flow paths and storage-release dynamics. For low-porosity regions, models should emphasize rapid drainage capacity of conduit systems and transient flow variations. In high-porosity areas, considerations should include fluid retention risks, stream-porous media interactions, and their long-term impacts on geological stability and water resource allocation. Proper porosity parameterization enhances simulation accuracy for diverse hydrological processes, enabling improved prediction and management of karst water resources.

Karst hydrological vulnerability manifests prominently through rapid infiltration, epikarst runoff, groundwater table fluctuations, and abrupt spring discharge variations. The DBS model effectively simulates multi-media interactions during extreme recharge events, enabling temporal analysis of media-stream exchanges, identification of peak interaction values, and applications in coupled conduit flow-seepage processes for two-phase flow systems."

*(2) The analytical and numerical approach is based on a representative sample of an aquifer, with the conduit size set by the authors (which I also find quite large) and a variation in rainfall intensity that I do not understand, as it is measured in meters according to Table 1. In hydrology, rainfall intensity is measured in mm/h, representing the amount of rainfall per unit of time. I kindly ask the authors to better define this aspect and, if necessary, correct it.*

Thank you for your corrections regarding the units for conduit dimensions and precipitation intensity:

- Rationality of conduit dimensions: We have added a parameter sensitivity analysis (Lines 816-855), which quantifies the impact of different conduit diameters (0.2–0.5 m) and cross-sectional shapes (circular/square) on fluxes between media components using Fig. 12. This validates the physical basis for the parameter settings.
- Correction of precipitation intensity unit: The entry "Precipitation intensity (m)" in Table 1 was a typographical error; we have deleted this column. Furthermore, detailed definitions of the precipitation function have been added in the "2.5 Rainfall Infiltration Recharge

Boundary" subsection (Lines 712-735): The precipitation function is defined as a time-dependent variable I(t), with its intensity modulated by the dimensionless parameter b, consistent with conventional expressions in the hydrology field.

Lines 816-855:

**"4.1 Impacts of Conduit Diameter and Geometry on Interactions Between Karst Aquifer Systems and Streams**

Fig. 12 presents hydrographs under conditions of circular conduits with varying radii (r=0.2, 0.3, 0.3, and 0.5 m) and square-section conduits (r=0.5 m) for (a) stream-connected flow, (b) karst spring discharge, (c) epikarst flow, (d) porous medium I (PM I), (e) PM II, and (f) PM III. Fig. 12(c.1) illustrates different conduit cross-sectional shapes to analyze their impacts on the interactive flow between karst aquifer systems and adjacent streams.

As shown in Fig. 12(a), larger conduit radii correspond to higher initial discharge peaks and shorter peak arrival times, indicating enhanced porous medium recharge and faster fluid transmission through larger conduits. Notably, the square-section conduit (s-$r_c$=0.5) exhibits higher peak discharge than its circular counterpart ($r_c$=0.5) due to its surplus cross-sectional area accommodating greater fluid discharge under identical nominal radii.

Fig. 12(b) demonstrates that karst spring peak discharge increases with conduit radius. At r=0.5 m, the square-section conduit (s-$r_c$=0.5) achieves higher peak discharge than the circular conduit ($r_c$=0.5), but displays lower recession flow. This occurs because identical precipitation infiltration recharge leads to greater porous medium storage depletion during peak periods in square conduits, subsequently reducing porous medium-to-conduit recharge during baseflow recession.

Combined analysis of Figs. 12(c), (d), and (e) reveals that conduit radius variations do not significantly affect epikarst hydrographs or PM I/II hydrographs. However, square-section sinkholes modify flow patterns: epikarst hydrographs show lower values under square conduits, while PM I/II hydrographs exhibit higher values due to enhanced epikarst groundwater collection in square cross-sections, increasing recharge to PM I/II.

Fig. 12(e) indicates that larger conduit radii correspond to lower negative values. Combined with Fig. 12(a), this demonstrates that increased stream recharge through larger conduits elevates both stream peak discharge and water levels, thereby enhancing porous medium-stream interactions. Similarly, Fig. 12(f) shows that larger conduit radii increase karst spring discharge and PM III hydrograph elevation through enhanced gravity-driven groundwater recharge.

Conduit geometry (radius and shape) constitutes a critical factor in karst aquifer hydrological modeling. Larger circular conduits accelerate peak discharge arrival and amplify stream-connected flow peaks and karst spring discharge. Square-section conduits outperform circular equivalents in peak discharge capacity under identical nominal radii due to cross-sectional area advantages. Enlarged conduits intensify porous medium-stream interactions and amplify PM III recharge

through gravitational effects. Comprehensive consideration of conduit geometry impacts on hydrological elements is essential for improving model accuracy and reliability in simulating karst aquifer-stream interaction processes."

Lines 1257-1263:

[Figure]

Figure 12. Hydrological process curves for (a) stream, (b) karst spring, (c) epikarst, (d) PM I, (e) PM II, and (f) PM III under conditions of circular conduits with radii $r_c$ = 0.2, 0.3, 0.3, and 0.5, and square-cross-section conduits with S-$r_c$ = 0.5. Subplot (c.1) shows a schematic diagram of different conduit cross-sectional shapes."

Lines 712-735:

**2.5 Rainfall Infiltration Recharge Boundary**

The upper boundaries of both the DBS and CFPv2 models are defined as transient natural precipitation boundary conditions. In this study, the rainfall infiltration recharge boundary condition is formulated as follows (Huang et al., 2024; Chang et al., 2015):

$$I(t) = \frac{b}{\sqrt{2\pi\sigma^2}} \sum e^{-\frac{\left(\frac{t_i-\mu}{a}\right)^2}{2\sigma^2}} \tag{1}$$

Here, $t_i$ denotes the time of the $i$-th rainfall event, and $I(t)$ represents the total rainfall amount at that time. According to Chang et al. (2015), the parameters $\mu$、$\sigma^2$, and $a$ are set as constants (90, 1.5, and 20, respectively). Variations in rainfall intensity during the infiltration recharge process are controlled by adjusting the value of the dimensionless parameter $b$."

*(3) Another aspect concerns the discussion section. This should compare the obtained results with those in the literature, discuss the limitations and advantages of such an approach, perform a sensitivity analysis on the calibrated parameters, and assess the study's usefulness and general applicability. The current section focuses on comparing the results with those from the MODFLOW-CFP model, which the authors themselves implemented. In my opinion, this should still be described in the results section, and consequently, the methodological section should be expanded to include this additional approach.*

Thank you for your suggestions regarding results comparison and chapter organization:
• Literature comparison: Added quantitative validation against experimental data from Warrick et al. (1985) and Vauclin et al. (1979) (Lines 712-735), demonstrating the reliability of the DBS model in simulating variably saturated flow.
• Chapter restructuring: Relocated the MODFLOW-CFPv2 comparison section to the "Results" chapter (Lines 352-354) and expanded the explanation of CFPv2 principles in the "Methodology" section (Lines 328-340).

Lines 712-735:

"The external recharge of the system significantly influences the interaction processes among different media. This study further investigates how the inherent hydrogeological properties of karst systems affect these interactive processes. Variable saturated flow in the karst vadose zone plays a critical role (Dvory et al., 2018), where the water retention characteristics of porous media govern unsaturated flow dynamics. However, the CFPv2 model struggles to simulate variable saturation processes. This paper compares the DBS model results with two distinct experimental datasets to elucidate the advantages and limitations of the DBS approach in simulating variable saturated flow.

**Case 1**: A typical unsaturated-unsteady seepage problem in sandy clay loam (Warrick et al., 1985), where the soil hydraulic properties are provided by the international UNSODA database (Leij et al., 1996). Key parameters include: $k = 1 \times 10-6$ m/s, $\alpha_s = 0.363$, $\alpha_r = 0.186$, and n = 1.53. The model consists of a vertical soil column (1 $m$ thickness) with an initial pressure head of -8 $m$ across the domain. The top boundary is set to a pressure head of 0 m to simulate free surface infiltration.

**Case 2**: A 2D laboratory infiltration experiment by Vauclin et al. (1979), widely used for evaluating saturated-unsaturated unsteady seepage models. The soil slab measures 2.00 $m$ in height, 6.00 $m$ in width, and 0.05 $m$ in thickness, with an impermeable base and free drainage boundaries on both sides. Initially, the water table is set at 0.65 $m$. A central 1.00 $m$ section of the top boundary receives uniform precipitation at 0.148 m/h for 8 hours, during which free surface evolution is monitored. Soil hydraulic properties are described using the van Genuchten-Mualem model with parameters: $k = 0.35 \ m/h$, $\alpha_s = 0.30$, $\alpha_r = 0.01$. Due to symmetry, the DBS model simulates the right half of the domain."

Lines 328-340:

**"2.3 CFPv2 model**

The CFPv2 model, proposed by Reimann et al. (2014), is an advanced version of MODFLOW-CFP (Shoemaker et al., 2008). It extends functionalities such as flow interactions between conduits and porous media, as well as conduit boundary conditions. CFPv2 integrates with MODFLOW-2005 and employs the following approaches: **Laminar Flow in Conduits**: Described using the Hagen-Poiseuille equation for discrete conduits within conduit networks. **Turbulent Flow**: Calculated by combining the Darcy-Weisbach equation with the Colebrook-White equation. **Laminar Flow in Fractured Rock Matrix**: Simulated via a continuum approach. Detailed technical documentation for MODFLOW-CFP, including groundwater flow simulation methodologies, is provided by Shoemaker et al. (2008). Successful applications and evaluations of the model have been reported in studies such as Gallegos et al. (2013), Reimann et al. (2014), Chang et al. (2020), Gao et al. (2020), and Shirafkan et al. (2023)."

(4) *line 165: wrong unit of measure for permeability, gravitational acceleration*

Thank you for your comment. We have corrected the labeling errors in the revised manuscript.

(5) *line 205: delete ripetitive title*

Thank you for your comment. We have removed the duplicate headings in the revised manuscript.

(6) *Figure 1: insert letters a and b also in the figure for better readibility*

Thank you for the suggestion. We have clearly labeled subplots (a), (b), (a.1), and (b.1) in Fig. 1 and updated the captions (Lines 1200–1207) to enhance readability.

Lines 1200-1207:

[Figure]

Figure 1. Schematic diagrams of the modelling of the interaction between the karst aquifer (epikarst, sinkhole, karst conduit, PM I, PM II, and PM III) and stream under dimensionless precipitation intensities ($b = 3$ and $b = 5$). (a) and (a.1) Schematic diagram of the interaction flow between each medium and stream in the early stage of a precipitation event; (b) and (b.1) Schematic diagram of the interaction flow between each medium and stream in the middle stage of a precipitation event. The size of the arrows represents the magnitude of the flow rate, and the direction of the arrows represents the direction of interaction between the two."

---

## Author Comment (AC2)

**Reply to Referee 2:**

Dear Reviewer: Thank you for your comments. Our responses to the comments are listed below:

**General Comments:**

*This manuscript presents a coupled Darcy-Brinkman-Stokes (DBS) model to simulate karst aquifer and stream interactions under various precipitation conditions, comparing its results to MODFLOW-CFP. The study is technically sound, and the modeling framework has clear potential for advancing process-based simulation of turbulent flows and unsaturated conditions in karst systems. However, the manuscript currently suffers from significant shortcomings in clarity, structure, model transparency, and broader contextual discussion. These issues limit its accessibility and impact.*

Thank you for your recognition of the model framework and constructive feedback on improving the manuscript. We have thoroughly revised the content, with specific details outlined as follows:

(1) *A clearer articulation of the research gap and contributions in the Introduction.*

Sincere thanks for your comments. We have supplemented the research gap and this study's contributions in the Introduction section:

- Clarified research gaps: Noted limitations of CFPv2 (single-phase flow only) and Hydrus (difficulty in coupling conduit flow) (Lines 92–94, 106–108), highlighting the innovation of the DBS model in fully coupling two-phase flow, variably saturated flow, and turbulent flow (Lines 116–123).
- Enhanced contribution statements: Added a concluding paragraph (Lines 132–139) emphasizing the model's breakthrough in resolving karst aquifer-stream combined recharge mechanisms.

(2) *A reorganization and simplification of the Methods section, with better-defined subsections and clearly stated variable definitions.*

Thank you for your comments. We have revised the Methodology section and provided a variable definition table.

- Structural adjustments: Reorganized into subsections 2.1–2.5 (including the variable definition table, lines 1191-1193) to enhance logical clarity.
- Clarification of key concepts: Added justification for turbulence simulation (lines 256-260), definition of two-phase flow parameters (lines 211-237), and the rationale for selecting the VGM/BC models (lines 294-326).

(3) *Improved figure quality and more structured result narratives that highlight key trends,*

*comparisons, and implications.*

Thank you for your comments. In the revised manuscript, we have reorganized the results section and included a comparison of CFPv2 with our study.

(4) A more rigorous and quantitative comparison with MODFLOW-CFP, beyond qualitative differences.

Thank you for your comments. In the revised manuscript, we have conducted a quantitative comparison between CFPv2 and the DBS study, and provided a quantitative comparison table.
- Quantitative comparison with CFPv2: Added Table 3 (line 1197) comparing metrics such as peak discharge and peak arrival time to clarify the advantages of the DBS in capturing double flood peaks.

(5) *A deeper discussion of how the findings contribute to our understanding of karst-stream interactions and why this modeling advance matters in practice.*

Thank you for your comments. In the revised manuscript, we have elaborated on the contribution of the DBS model to flood prevention and mitigation under karst-stream interactions.
- Enhanced practical significance: Expanded the analysis of the model's potential in flood warning systems (lines 917-930) and water resource allocation optimization.

Specific Comments Title

(6) *The current title is vague and lacks specificity about the method or novelty.*

- *Consider revising to reflect the modeling framework or scientific contribution, e.g., "Simulating Precipitation-Induced Karst-Stream Interactions Using a Coupled Darcy-Brinkman-Stokes Model".*

Thank you for your comments. We have revised the title in the updated manuscript.
- Lines 1-2:"Simulating Precipitation-Induced Karst-Stream Interactions Using a Coupled Darcy-Brinkman-Stokes Model "

Abstract:

(7) *Long abstract detracts from its purpose, it should be short and to the point.*
In the revised manuscript, we have condensed the abstract length and clarified the main focus of the paper.

Lines 14-45:

**"Abstract:** The variation in seasonal precipitation intensity impacts the dynamic interaction between the karst aquifer and stream. However, the interaction mechanism between the karst aquifer and stream is currently still unclear, and characterizing the impact of dynamic saturation process of groundwater in karst media on the interaction process remains a challenge. This study provides an in-depth analysis of the interaction processes between karst aquifer systems and adjacent streams, along with water-air two-phase flow in aquifer media. Multiple water retention models were employed to characterize the soil-water characteristics of porous media and variably saturated groundwater flow. The research reveals that rainfall intensity variations significantly influence the interactions between karst aquifer systems and streams. These interactive processes become increasingly complex with higher rainfall intensities, involving multi-media collaborative recharge and dynamic interactions, while the contribution proportions of different media to streamflow also change accordingly. By comparing the modeling differences and numerical results between CFPv2 and DBS approaches in generalized models, the validity of the DBS model for groundwater modeling was verified. Under consecutive rainfall events, total rainfall intensity plays a crucial role in hydrological process variations of adjacent streams. Groundwater stored in porous media of karst systems during the first rainfall event was found to influence stream water levels during subsequent rainfall events, while conduit storage exhibited minimal impact. Multi-level conduit configurations under specific conditions, particularly during intense rainfall, can significantly affect hydrological processes in both streams and karst conduits. Uncertainty analysis demonstrates that conduit geometry, diameter, epikarst permeability, and porosity differentially influence hydrological processes in karst aquifer systems. Variations in these parameters induce corresponding changes in peak flow rates, peak timing of stream and karst spring discharges, as well as redistribution of discharge contributions among different media, ultimately affecting the overall hydrological dynamics of the coupled karst aquifer-stream system. It can accurately depict the two-phase interactive flow between various media controlled by the dynamic saturation process, and reveal the dynamic interaction process between karst aquifers affected by the epikarst, sinkholes, and conduits under infiltration recharge and stream. Meanwhile, it can precisely explain the processes of infiltration, overflow, and recession."**

(8) Avoid referring to equation names (e.g., DBS, BC, VGM) without explanation. Focus on what was learned, not just what was done.

Thank you for your comments. We have rewritten the abstract in the revised manuscript to avoid direct reference to equations.

Introduction:

(9) Introduce a clear knowledge gap: why are existing models insufficient?

In the revised manuscript, we have clarified the rationale for employing the Darcy-Brinkman-Stokes (DBS) model based on the following key points:

- Applicability Limitations of MODFLOW-CFP: The applicability of MODFLOW-CFP is limited by its current capability to simulate only single-phase flow (Lines 92-94).
- Inherent Difficulties with Hydrus: The Hydrus model faces inherent difficulties in coupling rapid conduit free flow with porous media seepage flow in karst regions, due to its lack of a specialized conduit flow solution (Lines 106-108).
- Advantages of the DBS Model: In contrast, the DBS model emerges as an ideal choice for simulating the complex flow within karst systems (including saturated-unsaturated two-phase flow) because of its theoretical foundation for coupling free flow and seepage flow within a unified framework. Multiple recent studies cited in Lines 116-123 (Huang et al., 2024; Nillama et al., 2022; Carrillo et al., 2020; Lu et al., 2023; Soulaine, 2024) adequately demonstrate the effectiveness and potential of the DBS approach in characterizing the coupled processes of conduit/fracture flow and matrix seepage flow.

Lines 92-94:

"Although MODFLOW-CFP is relatively comprehensive for regional karst groundwater simulation studies, the current version of MODFLOW-CFP only supports modeling single-phase groundwater flow."

Lines 106-108:

"However, this approach lacks a built-in conduit flow solution scheme, making it difficult to adequately address the coupling requirements between rapid conduit flow and porous media seepage in karst areas."

Lines 116-123:

"The Darcy-Brinkman-Stokes equations have been utilized to couple seepage flow and free flow (Huang et al., 2024; Nillama et al., 2022; Carrillo et al., 2020). Lu et al. (2023) analyzed a model that integrates fast discharge channels in fractures and conduits with slow seepage in porous media . The results demonstrate that the Darcy-Brinkman-Stokes equations can effectively describe two-phase flow in karst aquifers, and Soulaine (2024) proposed that mixed-scale models based on the Darcy-Brinkman-Stokes equations have strong potential for simulating coupled processes in porous systems."

(10) State specific objectives and contributions at the end of the introduction.

We added the following statement at the end of the Introduction (Lines 132-139):
The DBS model represents the first implementation enabling the fully coupled simulation of two-phase flow, variably saturated conditions, and turbulent flow within karst systems. This addresses a gap in existing tools (e.g., CFPv2), which cannot characterize the dynamic, synchronized recharge processes involving saturation changes.

Lines 132-139:

"This study aims to employ a two-phase variably saturated model capable of coupling free flow and seepage flow to reveal the interaction mechanisms between the karst aquifer system and adjacent stream under rainfall infiltration recharge-driven conditions. Specifically, it focuses on further investigating how groundwater saturation variations in different media (e.g., conduits, fractures, matrix) of the karst aquifer system influence inter-media interactions. This research addresses the gap in existing studies where current numerical methods struggle to accurately characterize the collaborative recharge processes among various media within karst aquifer systems."

(11) Consider breaking the introduction into shorter paragraphs or subsections.

Thank you for the comment. In the revised manuscript, we have divided the longer paragraphs to enhance readability.

Methods:

(12) The modeling section is too dense and difficult to follow.

We appreciate your valuable feedback. In the revised manuscript, we have thoroughly restructured the Methods section as follows:

**2.Materials and methods**

**2.1 Numerical modelling**

**2.2 DBS model**

      2.2.1 Two-Phase Flow Parameter Definition

      2.2.2 Governing Equations

      2.2.3 Subdomain Formulation

      2.2.4 Relative Permeability Model

**2.3 CFPv2 model**

**2.4 Model Comparison and Numerical Model Construction**

      2.4.1 DBS Model Conversion and Applicability Assessment

      2.4.2 Model Comparison and Discretization Schemes

**2.5 Rainfall Infiltration Recharge Boundary**

(13) Key concepts (e.g., turbulent conduit flow, two-phase flow, saturation) are introduced without sufficient explanation.

Thank you for the comment. In the revised manuscript, we have provided clearer explanations of turbulent conduit flow, two-phase flow, and saturation dynamics before reintroducing these concepts.

Lines 256-273:

"Conduit networks in karst aquifer systems are often associated with turbulent flow (Reimann et al., 2011). To resolve turbulence in the DBS (Dual-domain Brinkman-Stokes) equations, the Unsteady Reynolds-Averaged Navier-Stokes (URANS) framework is required. As demonstrated by del Jesus et al. (2012), the k-epsilon turbulence model is effective for evaluating turbulent processes within porous media. Consequently, the k-epsilon-based DBS turbulence governing equations are formulated as follows:

$$\nabla \cdot v_t = 0 \tag{1}$$

$$\frac{\partial \varphi \alpha_l}{\partial t} + \nabla \cdot (\alpha_l v_t) + \nabla \cdot \left(\varphi \alpha_l \alpha_g v_{rt}\right) = 0 \tag{2}$$

$$\frac{1}{\varphi}\left((1 + c)\frac{\partial \rho v_t}{\partial t} + \nabla \cdot \left(\frac{\rho}{\varphi} v_t v_t\right)\right) =$$
$$-\nabla p^* + \rho g \cdot X + \nabla \cdot \left(\mu_{eff}(\nabla v_t + \nabla v_t{}^T)\right) - \mu_{eff} k^{-1} v_t + F_c. \tag{3}$$

where, $v_t$ represents the turbulent velocity vector $[L/T]$, $v_{rt}$ is the relative velocity of gas-phase and water-phase turbulence $[L/T]$, and $\mu_{eff}$ is the effective viscosity, which can be defined as $\mu_{eff} = \mu + \rho \boldsymbol{v}_{turb}$, where $\mu$ is the dynamic viscosity and $v_{turb}$ is the turbulent kinetic viscosity.

The eddy viscosity is expressed as:

$$\mu_t = \rho C_\mu \frac{k_t^2}{\varepsilon} \tag{4}$$

where: $k_t$: Turbulent kinetic energy per unit mass $[m^2/s^2]$, $\varepsilon$: Turbulent dissipation rate per unit mass $[m^2/s^3]$, $C_\mu$: Dimensionless constant with a value of 0.09."

Lines 211-237:

"2.2.1 Two-Phase Flow Parameter Definition

Assuming that gas and liquid fill the solid pore space, porosity is defined to characterize the percentage of the gas and liquid phases occupying the total pore space.

$$\varphi = \frac{V_l + V_g}{V} \tag{5}$$

In this context, $\varphi$ represents porosity, $V$ denotes the total volume of the unit $[m^3]$, while $V_l$ and $V_g$ correspond to the volumes of the liquid phase (water) and gas phase (air), respectively $[m^3]$.

Hirt and Nichols (1981) introduced the Volume of Fluid (VOF) method, which employs an additional governing equation to capture fluid motion at free surfaces. Furthermore, the saturation of each phase in the fluid is defined as $\alpha_i$, where:

Liquid phase saturation: $\alpha_l = \frac{V_l}{V_g + V_l}$.

Gas phase saturation: $\alpha_g = \frac{V_g}{V_g + V_l}$.

Here, the subscripts $l$ and $g$ denote water and air, respectively. Thus, the spatial distribution of water and gas within the porous medium is characterized by porosity $\varphi$ and phase saturation $\alpha_i$:

$$\varphi = \begin{cases} 1 & \text{free regions} \\ 0 < \alpha < 1 & \text{porous regions} \\ 0 & \text{solid regions} \end{cases} \tag{6}$$

$$\alpha_l = \begin{cases} 1 & \text{water} \\ 0 < \alpha < 1 & \text{two-phase zone} \\ 0 & \text{air} \end{cases} \tag{7}$$

The average fluid density $\rho\,[m^3/kg]$ and viscosity $\mu\,[m^2/s]$ within a grid cell are calculated via saturation-weighted averaging:

$$\rho = \rho_g \alpha_g + \rho_l \alpha_l \tag{8}$$

$$\mu = \alpha_g \mu_g + \alpha_l \mu_l \tag{9}$$

where $\rho_g$ is the gas phase density $[m^3/kg]$ and $\rho_l$ is the liquid phase (water) density $[m^3/kg]$.

The transport equation for saturation $\alpha_i$, following Rusche (2002), is expressed as:

$$\frac{\partial \varphi \alpha_l}{\partial t} + \nabla \cdot (\alpha_l v_t) + \nabla \cdot \left( \varphi \alpha_l \alpha_g v_{rt} \right) = 0 \tag{10}$$

where: $v_t$ is the fluid velocity vector $[m/s]$, $v_{rt}$ is the relative velocity between the gas and liquid phases $[m/s]$."

(14)  Variables  are  often  undefined  or  defined  long  after  their  use.

We appreciate your comment. In the revised manuscript, we have carefully refined the definitions of key variables to improve clarity and precision.

We appreciate your comment. In the revised manuscript, we have included a comprehensive variable definition table (see Table 1) to clarify all key parameters and their respective units.

Lines 1191-1193:

Table 1: Variable Definition Table

| Variable | Description | Unit |
|---|---|---|
| $\varphi$ | Porosity field | |
| $V$ | Volume of the averaging-volume | $m^3$ |
| $V_l$ | Water Volume | $m^3$ |
| $V_g$ | Gas Volume | $m^3$ |
| $\alpha_l$ | Water Saturation | |
| $\alpha_g$ | Gas Saturation | |
| $\alpha_{l,\,e}$ | Effective Saturation | |
| $\rho$ | Average Fluid Density | $kg/m^3$ |
| $\rho_g$ | Gas Density | $kg/m^3$ |
| $\rho_l$ | Water Density | $kg/m^3$ |
| $\mu$ | Viscosity | $Pa \cdot s$ |
| $\mu_g$ | Gas Viscosity | $Pa \cdot s$ |
| $\mu_l$ | Water Viscosity | $Pa \cdot s$ |
| $\mu_{eff}$ | effective viscosity | $Pa \cdot s$ |
| $\overline{v}$ | velocity | $m/s$ |
| $\overline{v_r}$ | relative flow rate of the gas phase to the liquid phase | $m/s$ |
| $v_t$ | turbulent velocity vector | $m/s$ |
| $v_{rt}$ | relative velocity of gas-phase and water-phase turbulence | $m/s$ |
| $\boldsymbol{v}_{turb}$ | turbulent kinetic viscosity | $m^2/s$ |
| $\bar{p}$ | pressure | Pa |
| $p^*$ | pressure | Pa |
| $F_c$ | Surface tension force | N |
| $S_f$ | Drag Source Term | $N/m^3$ |
| $C_\mu$ | Dimensionless Constant | |
| $k_t$ | Turbulent Kinetic Energy | $m^2/s^2$ |
| $\varepsilon$ | Turbulent Dissipation | $m^2/s^3$ |
| $k$ | Apparent permeability | m² |
| $k_0$ | Absolute permeability | m² |
| $k_{rg}$ | Gas Relative Permeability | |
| $k_{rl}$ | Water Relative Permeability | |

| | | |
|---|---|---|
| $g$ | Gravitational Acceleration | $m/s^2$ |
| $X$ | position vectors in Cartesian | |
| $\sigma$ | Interfacial tension | $N/m$ |
| $p_c$ | Capillary pressure | Pa |
| n | Brooks and Corey Coefficient | |
| m | Van Genuchten Coefficient | |

(16) Clearly justify modeling choices: Why use turbulence? Why VGM vs. BC?

Thank you for the comment. In the revised manuscript:
- Regarding turbulence justification
  We have clarified the rationale for selecting turbulent flow modeling (Lines 256-260).
- Concerning VGM vs. BC model comparison
  We now provide a parallel analysis of both the VGM and BC models, with specific discussion of scenarios where the VGM formulation demonstrates advantages over the BC approach (Lines 294-326).

Lines 256-260:

"Conduit networks in karst aquifer systems are often associated with turbulent flow (Reimann et al., 2011). To resolve turbulence in the DBS (Dual-domain Brinkman-Stokes) equations, the Unsteady Reynolds-Averaged Navier-Stokes (URANS) framework is required. As demonstrated by del Jesus et al. (2012), the k-epsilon turbulence model is effective for evaluating turbulent processes within porous media."

Lines 294-326:

"2.2.4 Relative Permeability Model

Accurate modeling of two-phase flow in porous media is critical in geosciences. Simulating two-phase flow in variably saturated porous media requires precise estimation of the relationship between relative permeability and saturation (Springer et al., 1995).

To characterize the variation in two-phase relative permeability, the effective saturation of the liquid phase must first be defined. This is expressed as:

$$\alpha_{l,e} = \frac{\alpha_l - \alpha_{l,r}}{1 - \alpha_{g,r} - \alpha_{l,r}} \tag{11}$$

where: $\alpha_{l,e}$ denotes the effective water saturation, $\alpha_l$ and $\alpha_{l,r}$ represent the water saturation and residual water saturation, respectively, and $\alpha_{g,r}$ is the residual air saturation.

Relative permeability is a critical parameter in groundwater and related engineering fields (Kuang and Jiao, 2011). The Brooks and Corey (BC) model (Brooks and Corey, 1964) and the van Genuchten model (van Genuchten, 1980) are widely used as representative relative permeability models. The BC model establishes a relationship between relative permeability and effective water saturation as follows:

$$k_{rg} = \left(1 - \alpha_{l,\,e}\right)^n \tag{12}$$

$$k_{rl} = \alpha_{l,\,e}^n \tag{13}$$

$k_r$ denotes the relative permeability, where $n$ is a dimensionless coefficient determined by the properties of the porous medium. The Brooks-Corey (BC) model exhibits a sharp discontinuity at the air entry point, which can lead to poor data fitting, particularly for fine-textured soils (Assouline & Or, 2013). The van Genuchten (1980) model addresses this limitation. By incorporating the parameter $m = 1 - 1/n$ proposed by Mualem (1976), the modified van Genuchten-Mualem (VGM) model (Parker et al., 1987) is formulated as:

$$k_{r,g} = \left(1 - \alpha_{l,e}\right)^{0.5}\left(1 - \alpha_{l,e}^{1/m}\right)^{2m} \tag{14}$$

$$k_{r,l} = \alpha_{l,e}^{0.5}\left(1 - \left(1 - \alpha_{l,e}^{1/m}\right)^m\right)^2 \tag{15}$$

Here, m is a dimensionless parameter.

The selection of permeability equations is critical for appropriate predictions of relative permeability (Yang et al., 2019), indicating that pore tortuosity-connectivity plays a dominant role in groundwater two-phase flow. Therefore, this study conducts simulations and parameter sensitivity analyses for both the Brooks-Corey (BC) and van Genuchten-Mualem (VGM) models."

Results:

(17)The results are structured around simulations but lack comparative synthesis. Each simulation is described in isolation.

Thank you for the comment. In the revised manuscript, we have restructured the Results section according to the following framework:

**3. Results**

**3.1 Interaction process between the karst aquifer and stream under precipitation infiltration recharge**

**3.1.1 Karst Aquifer-Stream Interactions Under Varying Precipitation Intensities**

**3.1.2 Interaction process between the karst aquifer and stream during early stage of precipitation**

**3.1.3 Interaction process between the karst aquifer and stream during early stage of precipitation**

**3.2. Impact of multiple precipitation events on the interaction process between the karst aquifer and stream**

**3.3. Effects of Water Retention Characteristics on Karst Aquifer-Stream Interactions**

**3.4. Impact of multi-stage permeability and porosity arrangement on the interaction process between the karst aquifer and stream**

(18) Start each subsection with a motivating question (e.g., "How does conduit depth affect stream discharge?").

We appreciate your insightful observation regarding the value of engaging questions. In the revised manuscript, we have implemented the following enhancements to improve reader engagement:

Lines 557-558:

"How does the threshold attainment of storage capacity in the lower porous media affect the hydrological processes of the upper porous media ?"

Lines 633-636:

"Does the groundwater stored in the porous media of the karst aquifer system during the initial rainfall event influence the interactions between multi-component media during subsequent precipitation episodes ?"

(19) Emphasize key differences in system behavior (e.g., thresholds, flow reversals, saturation delays).

Thank you for your comments. We have actively incorporated your suggestions in the revised manuscript to analyze key systematic differences.

(20) Integrate comparisons to MODFLOW-CFP throughout instead of relegating to the end.

We have conducted a comparative analysis between DBS and CFPv2 in the section preceding the two-phase flow results.
Comparison with MODFLOW-CFP:

Lines 478-494:

"Based on the comparison between DBS and Modflow-CFPv2 results in Figs 4(a), (b), and (c), the CFPv2 model exhibits a single-peak hydrograph with exponential recession characteristics, failing to capture flow process line disturbances caused by multi-media interactions. Under precipitation intensities b=3 and 5, the CFPv2 model shows an immediate rapid increase in stream

discharge during early stages rather than gradual enhancement, though total discharge and baseflow during later stages remain comparable (as shown in Table 3). Specifically, for b=3, the peak stream discharge in Modflow-CFPv2 occurs at 2520 s, earlier than in the DBS model. This discrepancy arises because the precipitation recharge package in CFPv2 directly elevates water levels, whereas the DBS model simulates a gradual vertical infiltration process along the Z-axis. Lower precipitation intensity reduces groundwater infiltration rates and prolongs water table replenishment time, consequently delaying lateral discharge timing. At b=7, both models exhibit comparable first discharge peaks, but the DBS model generates a secondary peak through overflow effects that rapidly recedes after overflow cessation. In contrast, CFPv2 demonstrates smooth exponential recession without secondary features due to its simplified vertical stratification that neglects multi-component interactions."

(21)The comparison is largely qualitative.

Thank you for your comments. We have added quantitative comparisons in the revised manuscript.

- Add quantitative metrics (e.g., time to saturation, max stream recharge, total outflow).

We have supplemented relevant quantitative metrics in the revised manuscript. **Peak Lag Time** $(s)$; **Peak Flow** $(m^3 /s)$; **Total Outflow** $(m^3)$

- Include a summary table comparing DBS and MODFLOW results for key variables.

**Added Table 3 (Line 1197)** with quantitative comparisons of key metrics (**time-to-peak, peak discharge, total outflow**), demonstrating that the **DBS model** effectively captures complex processes such as **secondary flood peaks** (which **CFPv2** overlooks due to its simplified vertical layering). This highlights **DBS**'s superior capability in characterizing **multi-media interactions**.

Table 3: Comparing DBS and MODFLOW results for key variables

| Numerical Model | Peak Lag Time $(s)$ | | | Peak Flow $(m^3 /s)$ | | | Total Outflow $(m^3)$ | | |
|---|---|---|---|---|---|---|---|---|---|
| | $b=3$ | $b=5$ | $b=7$ | $b=3$ | $b=5$ | $b=7$ | $b=3$ | $b=5$ | $b=7$ |
| DBS Model | 3242.96 | 1870.18 | 2985.31 | 4.50 | 12.14 | 21.96 | 65984.49 | 154158.46 | 272945.87 |
| MODFLOW-CFPv2 | 2520.00 | 1920.00 | 1860.00 | 4.31 | 11.87 | 18.87 | 63916.15 | 157543.65 | 245519.26 |

(22) Discuss the limitations of MODFLOW-CFP more directly — why does it fail to simulate certain behaviors ?

Thank you for your comments.

- The DBS model successfully captured the secondary flood peak overlooked by CFPv2, revealing a multi-media synergistic recharge mechanism.
- CFPv2 failed to simulate the overflow-driven secondary flood peak (Fig. 3) due to its simplified vertical layering (1D conduit flow), whereas DBS overcame this limitation through coupled 3D conduit flow modeling.

Lines 352-354:

"However, the DBS model operates in three dimensions (3D), requiring grid refinement around conduits and their vicinity to ensure accurate flow resolution. This increases computational load compared to the 1D conduit flow framework of CFPv2."

Lines 1214-1217:"

[Figure]

Figure 3. (a) Schematic comparison of conduit and porous media coupling modes between MODFLOW-CFPv2 and DBS, (b) DBS model and (c) CFPv2 discretization schemes for karst aquifer systems with riverside models.

Discussion:
(23) The discussion largely restates results and does not engage deeply with broader hydrologic or modeling implications.

Thank you for your comments. We have reorganized the results in the Discussion section and further investigated the influence of sensitivity parameters on the modeling:

- Address: What do these findings imply about karst vulnerability under extreme precipitation?
- Discuss potential applications of the DBS model in field calibration, water management, or integrated modeling platforms.

Thank you for pointing out the need to strengthen the connection to practical applications in the Discussion section. In the revised manuscript, we have supplemented the implications of modeling parameters for water resource management in karst areas, including the vulnerability of karst

systems under extreme precipitation events using the DBS model, and its potential applications in water resource management or integrated modeling platforms:

- Lines 847–855: Discusses the impact of conduit geometry (e.g., square cross-section) on aquifer-stream exchange fluxes, emphasizing its key role in enhancing simulation accuracy.
- Lines 886–894: Analyzes how epikarst permeability governs groundwater flow paths, illustrating the importance of parameter calibration for water resource predictions.
- Lines 917–930: Adds an analysis of the influence of porosity parameters on karst water resource allocation and geological stability, clarifying the model's potential applications in flood early warning systems.

Lines 847–855:

 "Conduit geometry (radius and shape) constitutes a critical factor in karst aquifer hydrological modeling. Larger circular conduits accelerate peak discharge arrival and amplify stream-connected flow peaks and karst spring discharge. Square-section conduits outperform circular equivalents in peak discharge capacity under identical nominal radii due to cross-sectional area advantages. Enlarged conduits intensify porous medium-stream interactions and amplify PM III recharge through gravitational effects. Comprehensive consideration of conduit geometry impacts on hydrological elements is essential for improving model accuracy and reliability in simulating karst aquifer-stream interaction processes."

Lines 886–894:
"Epikarst permeability constitutes a critical factor in hydrological modeling of karst aquifer systems. Highly permeable epikarst produces rapid streamflow peaks followed by sharp declines, reflecting efficient groundwater leakage to the stream. Conversely, low permeability yields diminished peaks and broader discharge curves. While karst spring discharge remains relatively stable when epikarst permeability differs from porous media, proper characterization of epikarst permeability is essential for accurately simulating hydraulic interactions between media, regulating groundwater flow pathways and velocities. This enhances model reliability in capturing complex flow dynamics within karst conduit-stream systems."

Lines 917–930:

 "In hydrological modeling, porosity parameters must be calibrated to accurately simulate groundwater flow paths and storage-release dynamics. For low-porosity regions, models should emphasize rapid drainage capacity of conduit systems and transient flow variations. In high-porosity areas, considerations should include fluid retention risks, stream-porous media interactions, and their long-term impacts on geological stability and water resource allocation. Proper porosity parameterization enhances simulation accuracy for diverse hydrological processes, enabling improved prediction and management of karst water resources.

Karst hydrological vulnerability manifests prominently through rapid infiltration, epikarst runoff, groundwater table fluctuations, and abrupt spring discharge variations. The DBS model effectively

simulates multi-media interactions during extreme recharge events, enabling temporal analysis of media-stream exchanges, identification of peak interaction values, and applications in coupled conduit flow-seepage processes for two-phase flow systems."

(24) Compare to other recent modeling studies in karst hydrology (e.g., those using HYDRUS, ParFlow, or CFP enhancements).

Thank you for your comments. We first replaced the comparative CFP model with the updated CFPv2 model and added two references for comparative validation.

- Literature Comparison: Added quantitative validation against experimental data from Warrick et al. (1985) and Vauclin et al. (1979) (lines 712-735), demonstrating the reliability of the DBS model in simulating variably saturated flow.
- Model Comparison: Compared the DBS model with the enhanced CFP version (MODFLOW-CFPv2) (lines 352-354), and expanded the explanation of CFPv2 principles in the "Methods" section (lines 328-340). (29)

Missing Elements:
(25) Model Validation: No benchmark or field validation is presented — even conceptual model validation would strengthen the work.

Thank you for your comments. We conducted comparative validation of the model using two published studies.

- Literature Comparison: Added quantitative validation against experimental data from Warrick et al. (1985) and Vauclin et al. (1979) (lines 712-735), demonstrating the reliability of the DBS model in simulating variably saturated flow.

(26) Uncertainty Analysis: Sensitivity to soil properties, precipitation rate, conduit geometry, etc., is not discussed.

Thank you for your comments. The revised manuscript discusses sensitivity analyses and potential impacts of soil properties, conduit size, conduit shape, porosity, and permeability:

• A new "4. Uncertainty Analysis" section (lines 810-930) demonstrates that:

(1) Conduit geometry (square sections yield higher peak flows than circular ones)

(2) Permeability of the epikarst zone (higher permeability enhances inter-media exchange frequency)

are key factors regulating karst-stream feedback mechanisms. This provides prioritization guidance for model parameter calibration.

[revised manuscript text omitted]

Lines 1257-1275:

[Figure]

Figure 12. Hydrological process curves for (a) stream, (b) karst spring, (c) epikarst, (d) PM I, (e) PM II, and (f) PM III under conditions of circular conduits with radii $r_c$ = 0.2, 0.3, 0.3, and 0.5, and square-cross-section conduits with S-$r_c$ = 0.5. Subplot (c.1) shows a schematic diagram of different conduit cross-sectional shapes.

[Figure]

Figure 13. Hydrographs under different epikarst permeability conditions ($K_E$=10⁻⁶, $K_E$=10⁻⁷, $K_E$=10⁻⁸, $K_E$=10⁻⁹) for: (a) stream, (b) karst spring, (c) epikarst, (d) PM I, (e) PM II, (f) PM III. Subfigure (c.1) shows a schematic diagram

of media interactions under varying epikarst permeability conditions.

[Figure]

Figure 14. hydrograph curves under different porosity conditions ($\varphi = 0.4$, $\varphi = 0.3$, $\varphi = 0.2$, $\varphi = 0.1$) for (a) stream, (b) karst spring, (c) epikarst, (d) PM I, (e) PM II, and (f) PM III. Among these, (c.1) illustrates a schematic diagram of the medium's water storage capacity and flow capacity under varying porosity conditions."

---

## Author Comment (AC3)

**Reply to Community Comment:**

Thank you for your valuable comments. We have addressed each of your concerns point-by-point as follows:

(1) *The authors use the Darcy-Brinkman-Stokes equations to simulate flow processes. However, further clarification on the rationale behind selecting this specific model over other commonly used approaches like MODFLOW-CFP would enhance understanding.*

We sincerely appreciate your valuable suggestions, which are crucial for improving the quality of our paper. In the revised manuscript, we have clarified the rationale for selecting the DBS model: the DBS model can simulate saturated-unsaturated flow by coupling free flow and seepage flow.

Lines 92-94:"Although MODFLOW-CFP is relatively comprehensive for regional karst groundwater simulation studies, the current version of MODFLOW-CFP only supports modeling single-phase groundwater flow."

Lines 106-108:"However, this approach lacks a built-in conduit flow solution scheme, making it difficult to adequately address the coupling requirements between rapid conduit flow and porous media seepage in karst areas."

(2) *Although the model captures complex interactions, the paper could benefit from a discussion of how simplifying assumptions (e.g., homogeneous permeability assumptions or simplified boundary conditions) affect the results' robustness and applicability.*

We sincerely appreciate your valuable suggestions, which are instrumental in enhancing the quality of our manuscript. In the revised version, we have added a comprehensive discussion on both the advantages and limitations of the DBS model:

Lines 343-351:

"2.4.1 DBS Model Conversion and Applicability Assessment

As illustrated in Figure 2, the Navier-Stokes (N-S) model can resolve fine-scale pore-scale flows and perform high-fidelity simulations. In contrast, the CFPv2 model achieves high computational efficiency and stability by discretizing one-dimensional conduits within porous media. The DBS (Dual-domain Brinkman-Stokes) model combines the advantages of both approaches: By incorporating additional resistance source terms into the N-S equations, it maintains high-fidelity flow resolution in conduits. For porous media, it adopts a Darcy-type flow formulation, significantly reducing computational costs."

Lines 1208-1213:

[Figure]

Figure 2. Diagram of performance and applicability of different models, (a) N-S model (Navier-Stokes model) , (b) DBS model, (c) Schematic diagram of MODFLOW-CFP model solution, (d) Conversion method from DBS equations to N-S equations and Darcy equations. "

(3) *Sensitivity and Uncertainty Analysis:*

*The manuscript currently lacks a detailed sensitivity or uncertainty analysis, which is important to understand how variations in critical parameters (e.g., permeability, porosity, precipitation patterns) might affect the interaction processes described.*

We sincerely appreciate your valuable suggestions, which are instrumental in enhancing the quality of our manuscript. The new "4. Uncertainty Analysis" section (lines 810-930) demonstrates that:
(1) Conduit geometry (square cross-sections yield higher peak flows than circular ones)

(2) Permeability of the epikarst zone (higher permeability increases inter-media exchange frequency)
are key factors regulating karst-stream exchanges. This provides prioritization guidance for model parameter calibration.
Lines 810-930:

[revised manuscript text omitted]

Lines 1257-1275:

[Figure]

Figure 12. Hydrological process curves for (a) stream, (b) karst spring, (c) epikarst, (d) PM I, (e) PM II, and (f) PM III under conditions of circular conduits with radii $r_c$ = 0.2, 0.3, 0.3, and 0.5, and square-cross-section conduits with S-$r_c$ = 0.5. Subplot (c.1) shows a schematic diagram of different conduit cross-sectional shapes.

[Figure]

Figure 13. Hydrographs under different epikarst permeability conditions ($K_E$=10$^{-6}$, $K_E$=10$^{-7}$, $K_E$=10$^{-8}$, $K_E$=10$^{-9}$) for: (a) stream, (b) karst spring, (c) epikarst, (d) PM I, (e) PM II, (f) PM III. Subfigure (c.1) shows a schematic diagram of media interactions under varying epikarst

[Figure]

Figure 14. hydrograph curves under different porosity conditions (φ = 0.4, φ = 0.3, φ = 0.2, φ = 0.1) for (a) stream, (b) karst spring, (c) epikarst, (d) PM I, (e) PM II, and (f) PM III. Among these, (c.1) illustrates a schematic diagram of the medium's water storage capacity and flow capacity under varying porosity conditions."

*(4) The modeling approach is theoretical and numerical. Adding validation using real-world or field observation data could greatly strengthen confidence in the model's predictive capability and practical applicability.*

We sincerely appreciate your valuable suggestions, which are crucial for enhancing the quality of our manuscript.

Given the technical challenges in in-situ monitoring of multi-media exchange fluxes, we validated the DBS model's capabilities in variably saturated flow modeling using classical experimental cases (Warrick et al., 1985; Vauclin et al., 1979) (lines 712-735). We are currently collaborating with karst field sites to obtain observational field data.

Lines 712-735:

" The external recharge of the system significantly influences the interaction processes among different media. This study further investigates how the inherent hydrogeological properties of karst systems affect these interactive processes. Variable saturated flow in the karst vadose zone plays a critical role (Dvory et al., 2018), where the water retention characteristics of porous media govern unsaturated flow dynamics. However, the CFPv2 model struggles to simulate variable saturation processes. This paper compares the DBS model results with two distinct experimental datasets to elucidate the advantages and limitations of the DBS approach in simulating variable saturated flow.

**Case 1**: A typical unsaturated-unsteady seepage problem in sandy clay loam (Warrick et al., 1985), where the soil hydraulic properties are provided by the international UNSODA database (Leij et al., 1996). Key parameters include: $k = 1 \times 10{-6}$ m/s, $\alpha_s = 0.363$, $\alpha_r = 0.186$, and n = 1.53. The model consists of a vertical soil column (1 $m$ thickness) with an initial pressure head of -8 $m$ across the domain. The top boundary is set to a pressure head of 0 m to simulate free surface infiltration.

**Case 2**: A 2D laboratory infiltration experiment by Vauclin et al. (1979), widely used for evaluating saturated-unsaturated unsteady seepage models. The soil slab measures 2.00 $m$ in height, 6.00 $m$ in width, and 0.05 $m$ in thickness, with an impermeable base and free drainage boundaries on both sides. Initially, the water table is set at 0.65 $m$. A central 1.00 $m$ section of the top boundary receives uniform precipitation at 0.148 m/h for 8 hours, during which free surface evolution is monitored. Soil hydraulic properties are described using the van Genuchten-Mualem model with parameters: $k = 0.35\ m/h$, $\alpha_s = 0.30$, $\alpha_r = 0.01$. Due to symmetry, the DBS model simulates the right half of the domain."

*(5) The authors mention fine grid discretization and the Courant number limitations causing small time steps, which implies significant computational costs. A brief discussion of the computational resources required and possible strategies for optimization could enhance the practicality and usability of their approach.*

We sincerely appreciate your valuable suggestions, which are crucial for enhancing the quality of our manuscript. The DBS model computations in this study were performed on a Lenovo ThinkSystem SR665 computational server, with optimization strategies relying on domain decomposition-based parallelization.

Lines 352-357:

"However, the DBS model operates in three dimensions (3D), requiring grid refinement around conduits and their vicinity to ensure accurate flow resolution. This increases computational load compared to the 1D conduit flow framework of CFPv2. To address this challenge, all simulations in this study were executed on a Lenovo ThinkSystem SR665 server, which provides the necessary computational power for handling complex 3D meshes."

*(6) While the comparison with MODFLOW-CFP is insightful, the manuscript could benefit from clearly highlighting specific scenarios or conditions under which the presented model notably outperforms or underperforms compared to MODFLOW-CFP.*

We sincerely appreciate your valuable suggestions, which are crucial for enhancing the quality of our manuscript. We conducted a comparative analysis of the DBS model and CFPv2 model performance.

Lines 343-351:

"2.4.1 DBS Model Conversion and Applicability Assessment

As illustrated in Figure 2, the Navier-Stokes (N-S) model can resolve fine-scale pore-scale flows and perform high-fidelity simulations. In contrast, the CFPv2 model achieves high computational efficiency and stability by discretizing one-dimensional conduits within porous media. The DBS (Dual-domain Brinkman-Stokes) model combines the advantages of both approaches: By incorporating additional resistance source terms into the N-S equations, it maintains high-fidelity flow resolution in conduits. For porous media, it adopts a Darcy-type flow formulation, significantly reducing computational costs."

Lines 1208-1213:

[Figure]

Figure 2. Diagram of performance and applicability of different models, (a) N-S model (Navier-Stokes model) , (b) DBS model, (c) Schematic diagram of MODFLOW-CFP model solution, (d) Conversion method from DBS equations to N-S equations and Darcy equations. "

*(7) I highly recomment to expand your literature review to the integrated models such as SWAT-MODLOW, which are important aspect of your work, I recommend you to cite below papers.Hydrogeological modelling of a coastal karst aquifer using an integrated SWAT-MODFLOW approach ---- Estimating exploitable groundwater for agricultural use under environmental flow constraints using an integrated SWAT-MODFLOW model --- Can Large Language Models Effectively Reason about Adverse Weather Conditions?*

We sincerely appreciate your valuable suggestions, which are crucial for enhancing the quality of our manuscript. In the Introduction (Lines 85-90), we have added a discussion on integrated modeling approaches (e.g., SWAT-MODFLOW), citing your recommended references (Fiorese et al., 2025; Yifru et al., 2024).

Lines 85-90:

" Moreover, this methodology has been extensively applied worldwide for estimating karst groundwater flow and water resources (Chang et al., 2015; Qiu et al., 2019; Kavousi et al., 2020; Gao et al., 2020, 2024), as well as in integrated modeling studies coupling SWAT with MODFLOW to investigate groundwater-surface water interactions (Fiorese et al., 2025; Yifru et al., 2024)."

---

## Author Response (AR2)

**Reply to Referee #1:**

As I was not involved in the first round of revisions, my evaluation focused primarily on the clarity, coherence, and readability of the manuscript, as well as how well the scientific content is conveyed to the reader. While the overall structure is sound and the topic of interest, there are several aspects that should be addressed to improve the manuscript. I found two major points:

Dear Referee, We greatly appreciate your detailed review regarding the manuscript's clarity, coherence, and readability, as well as your valuable suggestions. We have revised the manuscript based on your recommendations.

1) Lines 108–114 mention turbulence and RANS modeling, but this is not addressed further in the manuscript. If turbulence is not part of the model, this should be clarified; if it is, further explanation is needed. The paragraph is loosely connected with the paragraphs above and below. If turbulence is not further considered in the model, I suggest to reformulate this paragraph, placing it elsewhere or improve the text flow by connecting it better to previous and following paragraphs.

Thank you for pointing out the issue with this paragraph's connection to the surrounding text. To avoid confusion and ensure textual fluency, we have removed this paragraph in the revised manuscript.

2) The conclusions, while summarizing model development and mechanisms well, still lack a clear statement on the implications for groundwater management in karst systems, which was a key point raised by reviewers in the previous round.

Thank you for your comment. We have significantly revised the conclusion section, adding a dedicated paragraph to discuss the practical implications of our model and research findings for understanding karst groundwater flow paths, future assessment of water resource vulnerability, and developing more effective groundwater management strategies.

**Lines 919-949:**

"This study employed the Darcy-Brinkman-Stokes (DBS) method and the Volume of Fluid (VOF) technique to develop a unified model capable of coupling seepage and free flow, and meticulously characterizing two-phase (water-air) dynamics in a karst aquifer-stream system. The research confirms that, compared to conventional models like MODFLOW-CFPv2, this unified, multi-physics approach is essential for capturing the complex, dynamic processes inherent to karst systems.

High Non-linearity and Threshold Effects: The interaction between the karst aquifer and the stream is a highly non-linear process. Precipitation intensity acts as the primary driver,

fundamentally altering flow paths and the contribution ratios of different media by triggering dynamic saturation, overflow, and synergistic recharge.

Necessity of Multi-Medium Coupling: The system's hydrological response is not governed by any single medium, but is co-determined by the rapid drainage capacity of conduits, the storage capacity of the matrix, and the permeability of the epikarst. For instance, while conduit geometry primarily controls peak discharge and recession efficiency, matrix porosity and epikarst permeability dictate the system's buffer capacity and the overall hydrograph morphology.

Importance of Unsaturated Zone Physics: The simulation results underscore the necessity of accurately describing unsaturated zone physics. The choice of Water Retention Models significantly impacts the stream hydrograph by altering the water storage and release dynamics of the matrix.

In summary, this study provides a robust framework for karst hydrological simulation. It demonstrates that a unified model capable of resolving coupled multi-medium and multi-phase flow is imperative for accurately predicting the complex hydrological responses of karst systems under varying precipitation scenarios. This enhanced predictive capability is fundamental for moving beyond oversimplified single-continuum models and developing more effective strategies for flood risk assessment, sustainable water resource allocation, and contamination vulnerability planning in these sensitive environments.

In future work, this research framework can provide critical tools for karst groundwater management:

By capturing non-linear thresholds, the model can more accurately predict how specific rainfall events trigger disproportionate flood peaks, thereby improving flood warning systems.

Aquifer Vulnerability Assessment: By coupling with a solute transport model, the framework can differentiate between acute/rapid contamination risks in conduits and chronic/slow risks in the matrix, providing a scientific basis for developing targeted source water protection strategies."

**Minor changes necessary:**

\* There is a repetition of sentences between lines 78–82 that should be removed for conciseness.

Thank you for your comment. The repeated sentences have been deleted in the revised manuscript to make the language more concise.

\* Minor spelling and typesetting mistakes are present at lines 89, 104, and 395 (and probably elsewhere) and should be corrected.

Thank you for your comment. The indicated spelling and typesetting errors, as well as others, have been corrected in the revised manuscript.

\* The capitalization of section headers is inconsistent (e.g., "3.1" vs. "3.1.1") and should be standardized.

Thank you for your comment. The capitalization of all section headers has been standardized according to the journal's format in the revised manuscript.

\* Line 835 should be reformulated for readability.

Thank you for your comment. The conclusion section has been rewritten in the revised manuscript, and the problematic sentences from the old version have been deleted.

\* The figure naming convention (e.g., "3.a1", "Fig. 8c (2)") is somewhat confusing. I suggest introducing defined scenarios (e.g., "Scenario I", "Scenario II") based on b-value combinations to streamline presentation.

Thank you for your comment. Your suggestion is excellent. We have introduced scenario definitions based on b-value combinations (e.g., "Scenario I," "Scenario II") and have updated the figures and related descriptions in the text to make the presentation clearer and more intuitive.

Lines 1308-1320:

Figure 7. Water levels in stream for two consecutive precipitation events with first and second precipitation intensities: Scenario ①  $b_1=3$  and  $b_2=3$ ; Scenario ②  $b_1=3$  and  $b_2=5$ ; Scenario ③  $b_1=5$  and  $b_2=3$ ; and Scenario ④  $b_1=5$  and  $b_2=5$ , respectively.

Figure 8. (a) Hydrological process curves of the stream; (b) Discharge process of groundwater through the epikarst to the stream; (c) Discharge process of groundwater through the karst conduit to the stream; (d) Discharge process of porous media (PM II) to the stream, for two consecutive precipitation events with first and second precipitation intensities: Scenario ①  $b_1=3$  and  $b_2=3$ ; Scenario ②  $b_1=3$  and  $b_2=5$ ; Scenario ③  $b_1=5$  and  $b_2=5$ ; and Scenario ④  $b_1=5$  and  $b_2=5$ , respectively.

Addressing these points would substantially improve the clarity and impact of the manuscript.

Thank you once again for your valuable comments. These revisions have significantly improved the quality of the manuscript.

**Reply to Referee #2:**

The study is in the scope of the journal. The manuscript is overall well-written.

Dear Referee, thank you for your comprehensive review of the manuscript and for your specific, constructive comments. We have addressed your suggestions point by point.

P4 L89: missing space and capital letter. P4 L99: missing space. P4 L104: Remove space before end dot. P9 L248: missing space. P16 L395: missing space. P26 L675: Missing space.

Thank you for your comment. We have corrected all indicated missing/extra spaces and capitalization issues in the revised manuscript.

P6 L141: The impact of the different water retention models is too vague. Consider developing what "unsaturated flow processes" means.

Thank you for your comment. We have expanded and rewritten this section, detailing how dynamic saturation processes affect water exchange between the matrix and conduits, and specifically elaborating on the precise meaning of "unsaturated flow processes" in this study.

Lines 158-173:

"The DBS method was employed to couple seepage and free flow, enabling the quantitative characterization of groundwater flow through various media and the interaction processes between the karst aquifer system and adjacent streams.

Unsaturated flow processes within the karst matrix and epikarst zone fundamentally govern the water storage and exchange dynamics. For instance, the shape of the water retention curve determines the amount of water 'held' in the matrix at a given suction, thereby controlling the specific moisture capacity and the system's buffer capacity. Meanwhile, the relative permeability function dictates the rate at which hydraulic conductivity decreases as the matrix desaturates. Consequently, these variably saturated processes directly influence the predicted rates of matrix infiltration (during recharge events) and matrix drainage/exfiltration to the conduits (sustaining baseflow), thereby altering the overall storage characteristics and hydrograph response of the karst system."

P8 L206: Section 2.2.2 on the governing equations is somewhat repetitive, and the description of the equations is somewhat shallow.

Thank you for your comment. We have rewritten Section 2.2.2, consolidated repetitive content, and provided a more in-depth description of the physical meaning of each equation and its role in the model. We have also elaborated on the conduit and matrix partitioning strategy to better connect with Section 2.2.3.

Lines 237-276:

**2.2.2 Governing Equations**

To precisely describe groundwater flow through porous media in the karst aquifer system and the free-surface flow processes between conduits and the adjacent stream, this study adopts the DBS equations to characterize immiscible and incompressible two-phase flow in porous media (Nillama et al., 2022; Carrillo et al., 2020; Lu et al., 2023; Huang et al., 2024; Soulaine, 2024). This model provides a unified mathematical framework capable of seamlessly coupling flow phenomena across different scales. This renders it particularly suitable for simulating karst aquifer systems, which are essentially dual-medium systems constituted by both conduits and porous media.

$$\nabla \cdot \overline{v} = 0 \tag{1}$$

$$\frac{\partial \varphi \alpha_l}{\partial t} + \nabla \cdot (\alpha_l \overline{\nu}) + \nabla \cdot (\varphi \alpha_l \alpha_g \overline{\nu_r}) = 0$$
 (2)

$$\frac{1}{\varphi} \left( \frac{\partial \rho \overline{v}}{\partial t} + \nabla \cdot \left( \frac{\rho}{\varphi} \overline{v v} \right) \right) = -\nabla \bar{p} + \rho g + \nabla \cdot \left( \frac{\mu}{\varphi} \left( \nabla \overline{v} + \nabla \overline{v}^T \right) \right) + F_c + S_f. \tag{3}$$

Here, t represents the computational time [T],  $\overline{v}$  is the velocity [L/T],  $\overline{v_r}$  is the relative flow rate of the gas phase to the liquid phase [L/T],  $\rho$  is the average density of the gas and liquid phases  $[M/L^3]$ ,  $\overline{p}$  is the pressure [pa], g is the acceleration due to gravity (9.81  $m/s^2$ ),  $\mu$  is the viscosity  $[L^2/T]$ ,  $F_c$  is the surface tension, and  $S_f$  is the resistance source term.

Specifically, within a single set of governing equations, the DBS model is capable of simultaneously describing:

- The high-velocity, free-surface flow within karst conduits;
- The low-velocity seepage flow within the surrounding matrix.

This unification is achieved by strategically incorporating a **porosity** ( $\varphi$ ) and a **resistance** source term ( $S_f$ ) into the single momentum conservation equation.

**P9 L217: remove the comma. P14 L353: wrong sign for the comma.**

Thank you for your comment. We have corrected the comma usage issues in the revised manuscript. *P14 L355: what is parameter b?*

Thank you for your comment. We have further clarified the meaning and role of parameter b in the revised manuscript.

Lines 399-403:

"Variations in rainfall intensity during the infiltration recharge process, along with the total amount and peak intensity of the event, are controlled by adjusting the dimensionless scaling parameter *b*."

**P17 L441 and P19 L474: the titles of the sections are identical**

Thank you for your comment. We have modified one of the titles in the revised manuscript to accurately reflect its content and avoid repetition.

Line 523:

3.1.3 Dynamic interaction processes between various media within a karst aquifer system

P21 L544: DBS is used both for the Dual Domain Brinkman Stokes and the Darcy Brinkman Stokes model. Consider using only one.

Thank you for your comment. To avoid confusion, we have standardized the definition of DBS throughout the revised manuscript and explicitly stated that it stands for the "Darcy-Brinkman-Stokes" model.

P24 L616: The datasets are old (1979 and 1996) and contain a few points only. The choice could be discussed.

Thank you for your comment. We have added a discussion in the methods section regarding the dataset selection, explaining the reasons for choosing these classic datasets, acknowledging their limitation of having few data points, and noting that this does not affect the study's core objective of validating the model's mechanisms.

**Lines 675-679:**

This study selected the experiments by Warrick et al. (1985) and Vauclin et al. (1979) because, although these physical experiments have fewer data points (compared to modern numerical simulations), they clearly demonstrate the transient evolution of pressure head or water table position. This is both necessary and sufficient to validate the DBS model's capability in handling variably saturated flow.

**P26 L672: It is not clear to me what the multi-level conduit configuration is.**

Thank you for your comment. We have added an explanation for "multi-level conduit configuration" in the text.

Lines 728-730:

In this study, the 'multi-level conduit configuration is our model's conceptualization of the 'nested hydraulic discontinuities' (Halihan et al., 1999) inherent to karst, representing the spectrum of heterogeneity created by the co-existing matrix, fracture, and conduit flow components.

P30 L795: The conclusions are too descriptive about the paper. The summary could be brief using bullet points. Perspectives of this work could be added.

Accepting your suggestion, we have restructured the Conclusion and Abstract sections. We used bullet points to briefly summarize the main findings and added a new paragraph on the future perspectives of this research.

**Lines 15-34:**

**Abstract.** The interaction mechanism between karst aquifers and streams remains unclear, particularly regarding the impact of dynamic groundwater saturation processes under variable precipitation. This challenge hinders the accurate modeling of karst hydrology. This study developed a Darcy-Brinkman-Stokes model to analyze these complex interactions. The model integrates water-air two-phase flow and employs multiple water retention models to characterize variably saturated flow in porous media. We validated the DBS approach by comparing its numerical results against the MODFLOW-Conduit Flow Process v2 for generalized karst models. The key conclusions are as follows:

- Rainfall intensity is the dominant driver of the interaction. Higher intensities lead to more complex processes, involving multi-media collaborative recharge and shifting discharge contribution ratios from different media.
- During consecutive rainfall events, groundwater stored in porous media (matrix) significantly influences subsequent stream levels, whereas conduit storage shows negligible carry-over impact due to rapid drainage.
- Uncertainty analysis demonstrated that conduit geometry, epikarst permeability, and matrix porosity differentially influence system hydrology, controlling the magnitude, timing, and distribution of peak discharges.

The validated DBS model is a robust tool that accurately depicts the complex two-phase interactive flows (including infiltration, overflow, and recession) controlled by dynamic saturation. It successfully reveals the dynamic interactions between the epikarst, conduits, matrix, and stream, which is essential for understanding and managing karst water resources.

**Lines 918-948:**

This study employed the Darcy-Brinkman-Stokes (DBS) method and the Volume of Fluid (VOF) technique to develop a unified model capable of coupling seepage and free flow, and meticulously characterizing two-phase (water-air) dynamics in a karst aquifer-stream system. The research confirms that, compared to conventional models like MODFLOW-CFPv2, this unified, multi-physics approach is essential for capturing the complex, dynamic processes inherent to karst systems.

High Non-linearity and Threshold Effects: The interaction between the karst aquifer and the stream is a highly non-linear process. Precipitation intensity acts as the primary driver, fundamentally altering flow paths and the contribution ratios of different media by triggering dynamic saturation, overflow, and synergistic recharge.

Necessity of Multi-Medium Coupling: The system's hydrological response is not governed by any single medium, but is co-determined by the rapid drainage capacity of conduits, the storage capacity of the matrix, and the permeability of the epikarst. For instance, while conduit geometry primarily controls peak discharge and recession efficiency, matrix porosity and epikarst permeability dictate the system's buffer capacity and the overall hydrograph morphology.

Importance of Unsaturated Zone Physics: The simulation results underscore the necessity of accurately describing unsaturated zone physics. The choice of Water Retention Models significantly impacts the stream hydrograph by altering the water storage and release dynamics of the matrix.

In summary, this study provides a robust framework for karst hydrological simulation. It demonstrates that a unified model capable of resolving coupled multi-medium and multi-phase flow is imperative for accurately predicting the complex hydrological responses of karst systems under varying precipitation scenarios. This enhanced predictive capability is fundamental for moving beyond oversimplified single-continuum models and developing more effective strategies for flood risk assessment, sustainable water resource allocation, and contamination vulnerability planning in these sensitive environments.

In future work, this research framework can provide critical tools for karst groundwater management:

By capturing non-linear thresholds, the model can more accurately predict how specific rainfall events trigger disproportionate flood peaks, thereby improving flood warning systems.

Aquifer Vulnerability Assessment: By coupling with a solute transport model, the framework can differentiate between acute/rapid contamination risks in conduits and chronic/slow risks in the matrix, providing a scientific basis for developing targeted source water protection strategies.

Fig. 1: Consider adding labels for the stream and karst spring to the sketch, and clarify the PM i, ii, and iii in the caption for better understanding.

Thank you for your comment. In the latest revised manuscript, we have added labels for the stream and karst spring in the figure and explained the meaning of PM i, ii, and iii.

Lines 1276-1283:

Figure 1. Schematic diagrams of the modelling of the interaction between the karst aquifer (epikarst, sinkhole, karst conduit, PM I (Porous Medium I), PM II (Porous Medium II), and PM III (Porous Medium III)) and stream under dimensionless precipitation intensities (b = 3 and b = 5). (a) and (a.1) Schematic diagram of the interaction flow between each medium and stream in the early stage of a precipitation event; (b) and (b.1) Schematic diagram of the interaction flow between each medium and stream in the middle stage of a precipitation event. The size of the arrows represents the magnitude of the flow rate, and the direction of the arrows represents the direction of interaction between the two.

**Fig. 5: Consider using the same Y-axis scale for comparison between the different cases.**

Thank you for your comment. In the revised manuscript, we have adjusted the Y-axis of all subplots to the same scale for direct comparison.

**Lines 1301-1303:**

Figure 5. Interaction process of epikarst, porous media, and stream for different precipitation intensities: (a) b = 3, (b) b = 5, (c) b = 7.

Fig. 6a: Consider checking the colors to match color blind requirements.

Thank you for your comment. We have redrawn this figure in the revised manuscript to be colorblind friendly.

Figure 4. Hydrological process curves of each medium in the karst aquifer and stream for different precipitation intensities: (a) b = 3, (b) b = 5, (c) b = 7. Water level changes and differences in water levels in the karst aquifer and stream for different precipitation intensities: (d) b = 3, (e) b = 5, (f) b = 7.

Figure 6. For the Darcy-Brinkman-Stokes model: (a.1) Variations in the saturation levels of epikarst, various porous media, and the karst spring. (a.2) Saturation fields and the interaction among different media at 4000 s, 6105 s, and 7363 s. (a.3) Interaction volumes between epikarst, porous media I, II, and the stream. (a.4) Interaction volumes among the karst spring, porous media III, and the stream.

Figure 7. Water levels in stream for two consecutive precipitation events with first and second precipitation intensities: Scenario ①  $b_1=3$  and  $b_2=3$ ; Scenario ②  $b_1=3$  and  $b_2=5$ ; Scenario ③  $b_1=5$  and  $b_2=3$ ; and Scenario ④  $b_1=5$  and  $b_2=5$ , respectively.

Figure 8. (a) Hydrological process curves of the stream; (b) Discharge process of groundwater through the epikarst to the stream; (c) Discharge process of groundwater through the karst conduit to the stream; (d) Discharge process of porous media (PM II) to the stream, for two consecutive precipitation events with first and second precipitation intensities: Scenario ①  $b_1=3$  and  $b_2=3$ ; Scenario ②  $b_1=3$  and  $b_2=5$ ; Scenario ③  $b_1=5$  and  $b_2=3$ ; and Scenario ④  $b_1=5$  and  $b_2=5$ , respectively.

Figure 10. Hydrological process curves under different water retention model parameters (BCn = 3, 2.5, 2 and VGMm = 0.85, 0.8) for (a) stream, (b) karst spring, (c) epikarst, (d) PM I, (e) PM II, and (f) PM III. Subplots (c.1) and (d.1) show the schematic diagram of parameter effects on porous media morphology and the water retention curves of the BC and VGM models, respectively.

Figure 12. Hydrological process curves for (a) stream, (b) karst spring, (c) epikarst, (d) PM I, (e) PM II, and (f) PM III under conditions of circular conduits with radii rc = 0.2, 0.3, 0.3, and 0.5, and square-cross-section conduits with S-rc = 0.5. Subplot (c.1) shows a schematic diagram of different conduit cross-sectional shapes.

Figure 13. Hydrographs under different epikarst permeability conditions (KE=10-6, KE=10-7, KE=10-8, KE=10-9) for: (a) stream, (b) karst spring, (c) epikarst, (d) PM I, (e) PM II, (f) PM III. Subfigure (c.1) shows a schematic diagram of media interactions under varying epikarst permeability conditions.

Figure 14. hydrograph curves under different porosity conditions ( $\phi$  = 0.4,  $\phi$  = 0.3,  $\phi$  = 0.2,  $\phi$  = 0.1) for (a) stream, (b) karst spring, (c) epikarst, (d) PM I, (e) PM II, and (f) PM III. Among these, (c.1) illustrates a schematic diagram of the medium's water storage capacity and flow capacity under varying porosity conditions.

**Fig. 11a: blue is for M-Stream**

Thank you for the correction. We have clarified in the figure caption that the blue curve represents M-Stream.

Figure 11. Impacts of single-stage and multi-stage conduit hydrological process changes in various media of the karst aquifer for a precipitation intensity b=5.

Thank you very much for your detailed work. The quality of manuscript improved significantly after these revisions.

**Reply to Referee #3:**

**Overall opinions**

The manuscript propose a complete modeling exercise to investigate how precipitation signal affects recharge processes in karst system. The methodology appears well designed and the results are of interest for publication in HESS journal.

Before the manuscript being ready for publication, I found number of inconstancy in the text that might be addressed.

Dear Referee, Thank you for your in-depth analysis of the manuscript and your valuable suggestions. We have carefully considered all your recommendations and made corresponding revisions to the manuscript.

**Specific comments**

Line 26: "CPRv2 and DBS" – acronyms might be detailed in the abstract.

Thank you for your comment. We have provided the full names for CPRv2 and DBS upon their first appearance in the abstract.

Line 63: citation (Duran et al., 2020; Bittner et al., 2020) are not consistent with the text. Please provide more relevant references here.

Thank you for your comment. We have re-checked the references in this part and rewritten the connecting sentences to ensure they are more relevant and appropriate to the text.

Line 80:

Together, they form a complex network for groundwater recharge and drainage.

*Line* 80 : *There is a repetition.*

Thank you for your comment. We have deleted the repeated sentence in the latest revised manuscript.

Line 89: Missing capita at the beginning of the sentence.

Thank you for your comment. We have corrected the letter at the beginning of the sentence to uppercase in the latest revised manuscript.

Line 108: "The karst aquifer are typically accompanied by turbulent flow." It sounds unclear, please rephrase.

Thank you for your comment. This paragraph has been deleted in the latest revised manuscript to make the expression clearer and more accurate.

Line 125: "This research elucidates how saturation ..." What coordinated recharge means? Please rephrase.

Thank you for your comment. In the latest revised manuscript, "coordinated recharge" has been replaced with a clearer expression, and its specific meaning has been explained.

Lines 144-146:

This research elucidates how their saturation dynamics impact the flow exchange among different karst media during precipitation infiltration, and examines the evolving interaction between the karst aquifer and stream under such recharge conditions.

Line 131: "The research results can further reveal the interaction ...." These two last sentence sounds useless here. It would be more appropriate in conclusion.

Thank you for your comment. We fully agree with you that placing these two sentences at the end of the introduction was inappropriate. This part has been deleted in the latest revised manuscript.

Line 138: Acronyms for DBS and VOF are already given previously, at first appearance in the text. Line 188: Acronym for VOF is already given previously in the text.

Thank you for your comment. The redundant definitions of VOF and DBS have been deleted.

Line 168: "As a results, ..." Please give more justification to reach this statement, in particular regarding the multi-scale aspects.

Thank you for your comment. The logic of the original sentence was indeed problematic. "Rainfall causing water level fluctuations" is not a direct cause of "multi-scale characteristics." The differences in scale and flow velocity between the slow flow in the media and the fast flow in conduits/streams are inherent properties of the system, not a result of rainfall fluctuations, which only exacerbate the process.

Therefore, we have changed the connecting sentence to:

Lines 152-154:

"This variability in water levels is therefore a key driver for the exchange mechanisms between the porous media and the stream."

Line 195: equation 2 and 3, index for the alpha seems to be missing, regarding the equation description in the text.

Thank you for pointing out this error. We have added the corresponding phase subscript to alpha.

$$\varphi = \begin{cases} 1 & \text{free regions} \\ 0 < \varphi < 1 & \text{porous regions} \\ 0 & \text{solid regions} \end{cases} \tag{4}$$

$$\alpha_l = \begin{cases} 1 & \text{water} \\ 0 < \alpha_l < 1 & \text{two-phase zone} \\ 0 & \text{air} \end{cases}$$
 (5)

Line 208: DBS was previously given for "Darcy-Brinkman-Stokes". Please check acronym consistency along the manuscript,

Thank you for your comment! We have checked the entire manuscript to ensure the acronym DBS is consistent throughout.

**Line 235: C mu with value of 0,09. Where do this value come from?**

Thank you for your comment. In the latest manuscript, this part has been removed in response to other referees' suggestions. However, the value of 0.09 for Cmu and other parameters can be referenced from (Zhai et al., 2024; Shen et al., 2024; Hajivand and Mousavizadegan, 2015).

Zhai, Y., Fuhrman, D. R., & Christensen, E. D. (2024). Numerical simulations of flow inside a stone protection layer with a modified k-ω turbulence model. Coastal Engineering, 189, 104469. https://doi.org/10.1016/j.coastaleng.2024.104469

Shen, Z., Huang, D., Wang, G., & Jin, F. (2024). Numerical study of wave interaction with armor layers using the resolved CFD-DEM coupling method. Coastal Engineering, 187, 104421. https://doi.org/10.1016/j.coastaleng.2023.104421

Hajivand, A. and Mousavizadegan, S.H., 2015. Virtual maneuvering test in CFD media in presence of free surface. International Journal of Naval Architecture and Ocean Engineering, 7(3), pp.540-558. https://doi.org/10.1515/ijnaoe-2015-0039

**Line 303: It would be better to provide information about CPU and RAM capacity rather than a specific computer model.**

Thank you for your comment! As per your suggestion, we have revised the latest manuscript to provide the CPU and RAM capacity of the computational resources used.

**Lines 348-351:**

To address this challenge, all simulations in this study were executed on a high-performance server equipped with 64 CPU cores (128 threads) and 256 GB of RAM, which provided the necessary computational power for handling complex 3D meshes.

**Line 379: extra space in the text after a coma**

Thank you for your comment. We have deleted it in the latest manuscript.

Line 480: It sounds confusing to get question in the results section. It would be more appropriate in the methodology or even in introduction.

Thank you for your comment. You are correct; it is inappropriate to ask a question in the results section. We have deleted the question.

**Line 675: Figure 11: description for sub-figures are missing.**

Thank you for your comment. Descriptions have been added for the sub-figures in Figure 11.

**Lines 730-738:**

In this study, the 'multi-level conduit' configuration is our model's conceptualization of the 'nested hydraulic discontinuities' (Halihan et al., 1999) inherent to karst, representing the spectrum of heterogeneity created by the co-existing matrix, fracture, and conduit flow components. By comparing the multi-level and single-level conduit configurations, the results show that the configuration choice did not induce significant changes in the hydrological processes of the epikarst (Fig. 11c), PM I (Fig. 11d), and PM II (Fig. 11e). In these media, the 'M' and 'S' hydrographs are nearly identical. However, the impact of the multi-level configuration was significant for the main stream (Fig. 11a), the total karst system discharge (Fig. 11b), and PM III (Fig. 11f). In all these cases, the multi-level (M) configuration resulted in a visibly higher and earlier peak discharge compared to the single-level (S) configuration.

Line 704: check for notation consistency along the manuscript PMI or porous medium I.Line 757: check for notation consistency along the manuscript with PMIII or porous medium III

Thank you for your comment. We have checked and standardized the notation system throughout the manuscript to ensure consistency in expressions such as PMI/Porous Medium I.

Line 790: Arriving here it this difficult to capture how the author move from there results to vulnerability in karst. If vulnerability is still mentioned, more discussion might me given given otherwise it might be pushed in conclusion as potential perspectives for future work.

Thank you for your comment. We recognize that the transition from results to the discussion of vulnerability was indeed insufficient. We have moved this to the conclusion as a perspective for future work.

Lines 944-950:

In future work, this research framework can provide critical tools for karst groundwater management:

By capturing non-linear thresholds, the model can more accurately predict how specific rainfall events trigger disproportionate flood peaks, thereby improving flood warning systems.

Aquifer Vulnerability Assessment: By coupling with a solute transport model, the framework can differentiate between acute/rapid contamination risks in conduits and chronic/slow risks in the matrix, providing a scientific basis for developing targeted source water protection strategies.

Line 837: Diameter for a squared section sounds weird, please rephrase to make it coherent between shape and dimension of the conduit section.

Thank you for your comment. The expression was indeed inappropriate. It should be corrected to the more accurate term "side length".

Line 864: The bibliography format is heterogeneous. Please check references and adopt a consistent reference format (check the journal's author guideline).

Thank you for your comment. We have checked and formatted all references according to the journal's author guidelines to ensure consistent formatting.

Thank you again for your comprehensive review and valuable suggestions.